# Longitudinal deep multi-omics profiling in a *CLN3*^Δex7/8 minipig model identifies biomarker signatures of disease
Mitchell J. Rechtzigel [1], Brittany Lee[2], Christine Neville [3], Ting Huang [2], Ramón Díaz[4], Alex Rosa Campos [4], Khatereh Motamedchaboki[2], Daniel Hornburg [2], Tyler B. Johnson[1], Vicki J. Swier[1], Jill M. Weimer [1,3,5] & Jon J. Brudvig [1,3,5] ✉

## Abstract

**Background** Development of therapies for CLN3 disease, a rare pediatric lysosomal storage disorder, has been hindered by the lack of etiological insights and translatable biomarkers to clinics.

**Methods** We used a deep multi-omics approach to discover blood-based biomarkers using longitudinal serum samples from a porcine model of CLN3 disease. Comprehensive metabolomics was combined with a nanoparticle-based LC-MS-based proteomic profiling coupled with TMTpro 18-plex to generate quantitative data on 769 metabolites and 2634 proteins, collectively the most exhaustive multi-omics profile conducted on serum from a porcine model. This was previously impossible due to lack of efficient deep serum proteome profiling technologies compatible with model organisms.

**Results** Here we show that the presymptomatic disease state is characterized by elevations in glycerophosphodiester species and lysosomal proteases, while later timepoints are enriched with species involved in immune cell activation and sphingolipid metabolism. Cathepsin S (CTSS), Cathepsin B (CTSB), glycerophosphoinositol, and glycerophosphoethanolamine captured a large portion of the genotype-correlated variation between healthy and diseased animals, suggesting that an index score based on these analytes could have great utility in the clinic.

**Conclusions** This study's findings demonstrate the potential of deep multi-omics profiling for uncovering disease-specific biomarkers, providing valuable insights for understanding disease and facilitating the identification of potential drug targets, thus offering valuable insights for therapeutic interventions.

## Plain language summary

Batten disease is a rare childhood disorder in which brain cells become damaged and there is a decline in physical and mental abilities. It is difficult to monitor disease progression or response to treatments as there are no established components of the blood (biomarkers) that can be used to identify people with the disease. Identifying such biomarkers is critical to understanding why disease develops and to enable development of treatments. We used a large animal model of Batten disease to investigate whether blood-based biomarkers could be identified. We developed a scoring framework that accurately distinguishes disease from control samples. These findings suggest that the identified biomarkers have the potential to be used to determine response in clinical trials.

Batten Disease (also known as neuronal ceroid lipofuscinoses (NCLs)), are a group of neurodegenerative lysosomal storage disorders that result from pathogenic variants in one of 13 ceroid lipofuscinosis neuronal (CLN) genes. Collectively, Batten disease affects approximately 1 in 100,000 individuals worldwide, making it the most common pediatric neurodegenerative disorder[1]. The most common form of Batten Disease, CLN3 disease, is a rare and fatal autosomal recessive disorder caused by mutations in *CLN3*. Individuals with CLN3 disease typically experience vision loss in early childhood, followed by seizures, motor and cognitive decline, and

premature death by the third decade of life[2,3]. Pathologically, CLN3 dysfunction cascades from the accumulation of lysosomal storage material, microglia and astrocyte activation, to neuronal dysfunction and death[4].

Despite many years of research, the molecular function of CLN3 and many of the other NCL proteins has yet to be fully elucidated. Recent advancements have begun to outline a function for the CLN3 protein, but the field still lacks robust biomarker signatures that comprehensively reflect the disease state[5,6]. With the growing list of CLN3 disease-specific therapies entering clinical trials, there is a substantial need for non-invasive

[1]Pediatrics and Rare Diseases Group, Sanford Research, Sioux Falls, SD, USA. [2]Seer, Inc., Redwood City, CA, USA. [3]Discovery Science, Amicus Therapeutics, Philadelphia, PA, USA. [4]Sanford Burnham Prebys, La Jolla, CA, USA. [5]Department of Pediatrics, University of South Dakota Sanford School of Medicine, Vermillion, SD, USA. ✉e-mail: jbrudvig@amicusrx.com

biomarkers that can track disease progression and therapeutic efficacy[2,7]. We recently identified a group of glycerophosphodiesters (GPDs) as promising blood-based biomarker candidates for CLN3 disease[8]. Shortly thereafter, it was demonstrated that CLN3 is required for the clearance of GPDs from lysosomes[6]. Recent work also demonstrated that closely related phosphoinosides mediate lysosomal repair, suggesting that disrupted GPD metabolism or transport could underlie the severe lysosomal dysfunction that characterizes cellular disease pathology[9]. Although elevation of these GPD species closely corresponds with the absence of functional CLN3, this phenotype does not correlate with other progressive aspects of disease progression such as neuroinflammation and neurodegeneration. In contrast, markers of neurodegeneration such as neurofilament light (NFL) show highly variable elevations (2096 ± 1202 pg/mL) in CLN3 disease and overlapping data spread with healthy controls, and thus have questionable utility as diagnostic and prognostic biomarkers[10]. Overall, it is unlikely that any single biomarker will be adequate to evaluate the intricate nature of disease status, longitudinal progression, and response to therapy in an individual patient. Additionally, clinical trials are often reliant on placebo controls or natural history studies. A combined biomarker score that integrates diverse sets of markers reflecting different facets of disease etiology and pathology could provide powerful insights and tracking tools to accelerate drug development in ways that natural history studies cannot.

We seek to uncover a more diverse set of CLN3 disease-related biomarkers and gain insights into the molecular function of CLN3 using an untargeted metabolomics and a deep multi-nanoparticle-based proteomics in our recently described Yucatan minipig model of CLN3 disease harboring the most common patient mutation: a ~1 kb deletion in exons seven and eight[11,12] (Fig. 1). Here, we analyze blood serum samples from the model animals at pre-symptomatic (6-month), mid-stage/symptomatic (24-month), and late-stage (36-month) stages of disease progression to capture temporal changes and disease-specific patterns in the proteome and metabolome at both the pathway and individual molecule levels. In contrast to tissue biopsies, sampling blood can serve as a minimally invasive procedure for monitoring disease progression, enabling comprehensive proteomic research and in-life monitoring of biomarker status. However, historically the extreme dynamic range of blood protein concentrations has required a trade-off between depth of unbiased proteome coverage and number of samples analyzed, in particular for model organisms that lack abundant protein depletion solutions and that are not compatible with targeted strategies based on aptamers and antibodies designed for human proteomes[13]. Here, we utilize a nanoparticle-based protein sampling technology, the Proteograph™ Product Suite (Seer, Inc.), enabling deep and scalable proteomic profiling, independent of disease model species[11,14,15].

Our in-depth profiling of thousands of proteins with more than 10,000 peptides, integrated with in-depth metabolome data, uncovers unique signatures and biomarker candidates for CLN3 disease and provides insights into perturbed pathways in the CLN3 disease state.

## Methods

### Pig biofluid collection
Male and Female transgenic $CLN3^{\Delta ex7-8}$ Yucatan minipigs were generated as previously described[8]. Briefly, pigs were anesthetized with xylazine (TKX) and isoflurane (1–2%) for terminal tissue collection. Briefly, a 16 G needle attached to a 20-cc syringe was inserted into the right ventricle of the heart, and ~20 mL of blood was drawn. A Saf-T Holder™ transfer device was used to expel collected blood into two 10 mL Monoject™ blood collection tubes. Blood samples were placed at room temperature for 30 min and allowed to clot, after which they were centrifuged at $3100 \times g$ for 10 min at room temperature. Serum was then collected into 2 mL polypropylene screw-top tubes and stored at −80 °C. Samples were collected at 6-months ±7 weeks (pre-symptomatic), 24-months ±10 weeks (mid-stage/symptomatic), 36-months ± 6 weeks (late-stage), and 48-months ± 12 weeks (end point) in both $CLN3^{\Delta ex7-8}$ ($n = 6$ (3M/3F); 9 (6M/3F); 9 (5M/4F); 3 (2M/1F); respectively) and control ($n = 6$ (3M/3F); 9 (5M/4F); 9 (6M/3F); 3 (3M); respectively) mini pigs (Fig. 1), reflecting timepoints chosen for a parallel animal model characterization study. Wild type (WT) and transgenic $CLN3^{\Delta ex7-8}$ Yucatan miniature pigs were housed and maintained at Exemplar Genetics under an Exemplar Genetics approved Institutional Animal Care and Use Committee (IACUC) protocol (MRP2015-005). We affirm that this study has received ethical approval and that we have complied with all relevant ethical regulations for animal testing.

### Untargeted metabolomics discovery
**Metabolomic sample preparation.** Metabolomic analyses were performed by Metabolon (Morrisville, North Carolina, USA) as previously described[8]. Briefly, samples were shipped on dry ice overnight to Metabolon. Samples were inventoried and promptly stored at −80 °C until analyzed. Automated sample preparation was conducted using the MicroLab STAR® system (Hamilton Company) utilizing several recovery QC standards preceding the extraction process. Proteins were precipitated with methanol in a Genogrinder 2000 (Glen Mills) followed by centrifugation. Samples were divided into five fractions (two aliquots for replicate analyses with Reversed Phase Ultrahigh Performance Liquid Chromatography coupled to Mass Spectrometry (RP)/UPLC-MS/MS using positive ion mode electrospray ionization (ESI), one aliquot for RP/UPLC-MS/MS, with negative ion mode ESI, one aliquot for HILIC/

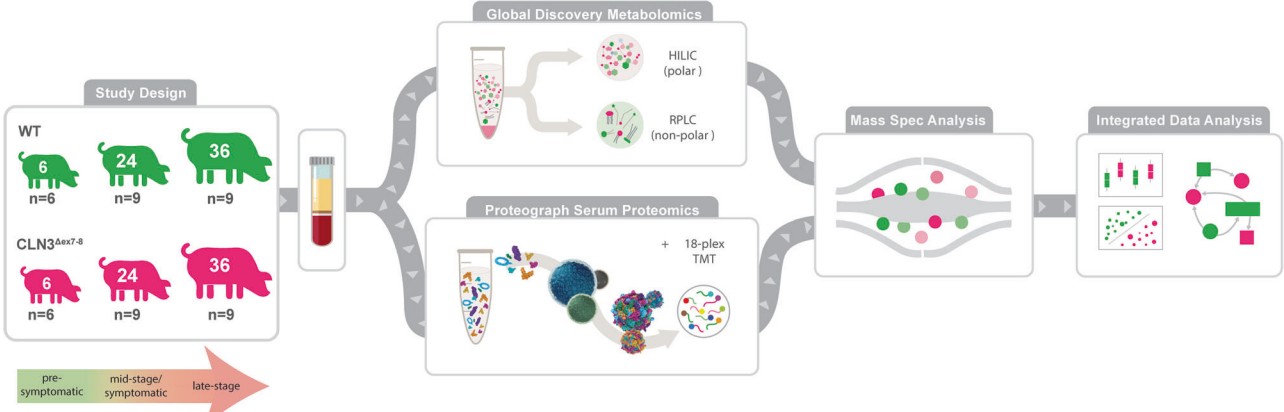

**Fig. 1 | Study design and multiomics analysis workflow.** Serum samples were taken from control and $CLN3^{\Delta ex7-8}$ pigs at 6-, 24-, and 36-months ($n = 6$, 9, and 9, respectively). One aliquot of serum was analyzed with the Global Discovery Panel: a comprehensive mass spectrometry analysis including both metabolites and lipids.

Another aliquot of serum was used to perform deep proteomics using Seer's nanoparticle technology and labeling with Tandem Mass Tag, high throughput mass spectrometry method. After mass spectrometry analysis of each omic data, the data was processed and integrated for statistical, multivariate analyses.

UPLC-MS/MS in negative ion mode ESI, and one sample aliquot was reserved for backup). The organic solvent was removed via TurboVap® (Zymark). Samples were stored overnight in liquid nitrogen prior to LC-MS/MS analysis.

**Quality assurance for metabolomic analysis.** To monitor liquid chromatography-mass spectrometry (LC-MS) instrument performance and chromatographic alignment, internal controls were included with experimental samples for each run. As previously described, these included pooled matrix controls of well-characterized human serum, process blanks consisting of extracted water samples, and a cocktail of QC standards that were selected not to interfere with the measurement of endogenous compounds[8,16,17]. Median relative standard deviation (RSD) was calculated for the QC standards to monitor instrument variability from run to run. Median RSD was calculated for all endogenous metabolites (i.e., non-instrument standards) present in 100% of the pooled matrix samples to account for overall process variability. QC samples were spaced evenly among the injections with experimental samples randomized across the platform run.

**Ultrahigh performance liquid chromatography-tandem mass spectroscopy analysis for discovery metabolomics.** As previously described, ultra-performance liquid chromatography coupled to tandem mass spectrometry (UPLC-MS/MS) analysis was conducted on a Waters™ ACQUITY™ UPLC and a Thermo Fisher Scientific™ Q Exactive™ Orbitrap™ high resolution and accurate mass spectrometer interfaced with a heated electrospray ionization (HESI-II) source and the Orbitrap mass analyzer operated at 35,000 mass resolution[8]. Samples were dried and reconstituted in method-compatible buffers for each of the LC-MS/MS acquisitions, which contained internal standards at fixed concentrations to control for injection and chromatographic run variations. For LC-MS/MS runs conducted in acidic positive ion conditions, chromatographically optimized for more hydrophilic compounds, the extracts were applied to a C18 column (Waters UPLC BEH C18-2.1 × 100 mm, 1.7 μm) followed by isocratic elution using water and methanol, containing 0.05% perfluoropentanoic acid (PFPA) and 0.1% formic acid (FA). Additionally, a separate aliquot was analyzed using acidic positive ion conditions; however, the extract was gradient eluted from the same C18 column using methanol, acetonitrile, water, 0.05% PFPA and 0.01% FA and was operated at an overall higher organic content. For samples analyzed using basic negative ion optimized conditions, a separate dedicated C18 column was utilized, and extracts were gradient eluted using methanol and water, however, with 6.5 mM ammonium bicarbonate at pH 8. The fourth aliquot was analyzed via negative ionization following elution from a hydrophilic interaction liquid chromatography (HILIC) column (Waters UPLC BEH Amide 2.1 × 150 mm, 1.7 μm) using a gradient consisting of water and acetonitrile with 10 mM ammonium formate, pH 10.8. The MS analysis alternated between MS and data-dependent MS[n] scans using dynamic exclusion. The scan range covered 70–1000 *m/z*, varying slightly across methods. Raw data files were extracted and analyzed as described in the data analysis section and have been deposited to the MetaboLights data repository[18].

**Metabolomic data extraction and compound identification and quantification.** Metabolon's hardware and software systems were used for raw data extraction, peak identification, and quality control processing. Compounds were identified by comparing their retention time/index (RI), mass to charge ratio (*m/z*), and tandem MS/MS spectral data matched to library entries of purified standards or recurrent unknown entities. Metabolites were quantified using area-under-the-curve.

**Metabolomics data processing.** To estimate relative abundances, metabolite chromatographic peak area data was $\log_2$-transformed, then values for each individual metabolite were scaled across samples to a mean of zero and unit variance. No batch normalization was necessary as all metabolites were detected in one run. Metabolites missing in more than 50% of the samples were removed from consideration, leaving 769 metabolites in the dataset. We present an in-depth analysis of data used in our previous work, however, comparisons and analyses made here are, to the best of our knowledge, novel[8].

## Targeted protein analysis
Neurology 4-PlexA targeted proteomic analysis was performed in singlet at a 4:1 sample dilution at the Simoa® Accelerator Laboratory (Billerica, Massachusetts, USA). This targeted proteomics panel included data for four proteins (GFAP, NFL, TAU, and UCHL1) and was processed in the same manner as the metabolomics data (Supplementary Fig. 1).

## Deep discovery proteomics analysis
**Automated blood serum sample processing with proteograph™ workflow.** Raw serum samples were processed by SP100 automation instrument with Proteograph™ Assay Kt included in the Proteograph Product Suite (Seer, Inc.) using five distinctly functionalized nanoparticles (NPs). In the fully automated workflow, 250 μL of serum was equally aliquoted into 5 tubes where 40 μL of serum from each tube was incubated with functionalized NPs included in the Proteograph Assay Kit. A 1-h incubation with NP surfaces allowed for protein corona formation to reach equilibrium and was followed by a series of gentle washes using the super-paramagnetic properties of the NPs to remove non-specific and weakly bound proteins.

Proteins bound to the NPs were then reduced, alkylated, and digested with Trypsin/Lys-C to generate tryptic peptides for downstream LC-MS/MS analysis. All steps were performed in a one-pot reaction directly on the NPs. The in-solution digestion mixture was then desalted, and all detergents were removed using a mixed media filter plate and a positive pressure (MPE) system.

Clean peptides were then eluted in a high-organic buffer into a deep-well collection plate. Immediately after peptide elution, peptide quantitation assay was performed using the Pierce Fluorescent Assay Kit to determine the peptide yield for each well.

**Sample multiplexing with TMTpro 18-plex labeling.** As shown in Fig. 2a, NP peptides were pooled, dried in a SpeedVac (3 h), and then reconstituted directly in 50% acetonitrile in 100 mM HEPES (pH 8) containing one of the TMT tags from the TMTpro 18-plex reagent (Thermo Fisher Scientific). The peptide-TMT mixture was incubated in a thermomixer for 1 h at 25 °C and 600 rpm, and the reaction was stopped by addition of 2% hydroxylamine to a final concentration of 0.2% and incubated for 15 min at 25 °C and 600 rpm. Labeled peptides for each 18-plex batch were pooled and dried using a SpeedVac system and subsequently reconstituted in 0.1% FA for desalting using a C18 TopTip (PolyLC, Columbia, Maryland) according to the manufacturer's recommendation.

**Two dimensional LC-MS/MS proteomics analysis.** Desalted TMT-labeled peptide pools were dried in a SpeedVac system and reconstituted in 20 mM ammonium formate pH ~10 for chromatography fractionation using a Waters Acquity BEH C18 column (2.1 × 15 cm, 1.7 μm pore size) mounted on an M-Class UPLC system (Waters). Peptides were then separated using a 35-min gradient: 5% to 18% B in 3 min, 18% to 36% B in 20 min, 36% to 46% B in 2 min, 46% to 60% B in 5 min, and 60% to 70% B in 5 min ($A = 20$ mM ammonium formate, pH 10; $B = 100\%$ ACN). A total of 72 fractions were collected and pooled in a non-contiguous manner into 36 total fractions. Pooled fractions were dried to completeness in a SpeedVac concentrator prior to mass spectrometry analysis.

Dried peptide fractions were reconstituted with 2% ACN, 0.1% FA and analyzed by Reversed Phase (RP) LC-MS/MS using an EASY-nLC™ 1200 system (Thermo Fisher Scientific) coupled to an Orbitrap LumosTribrid™

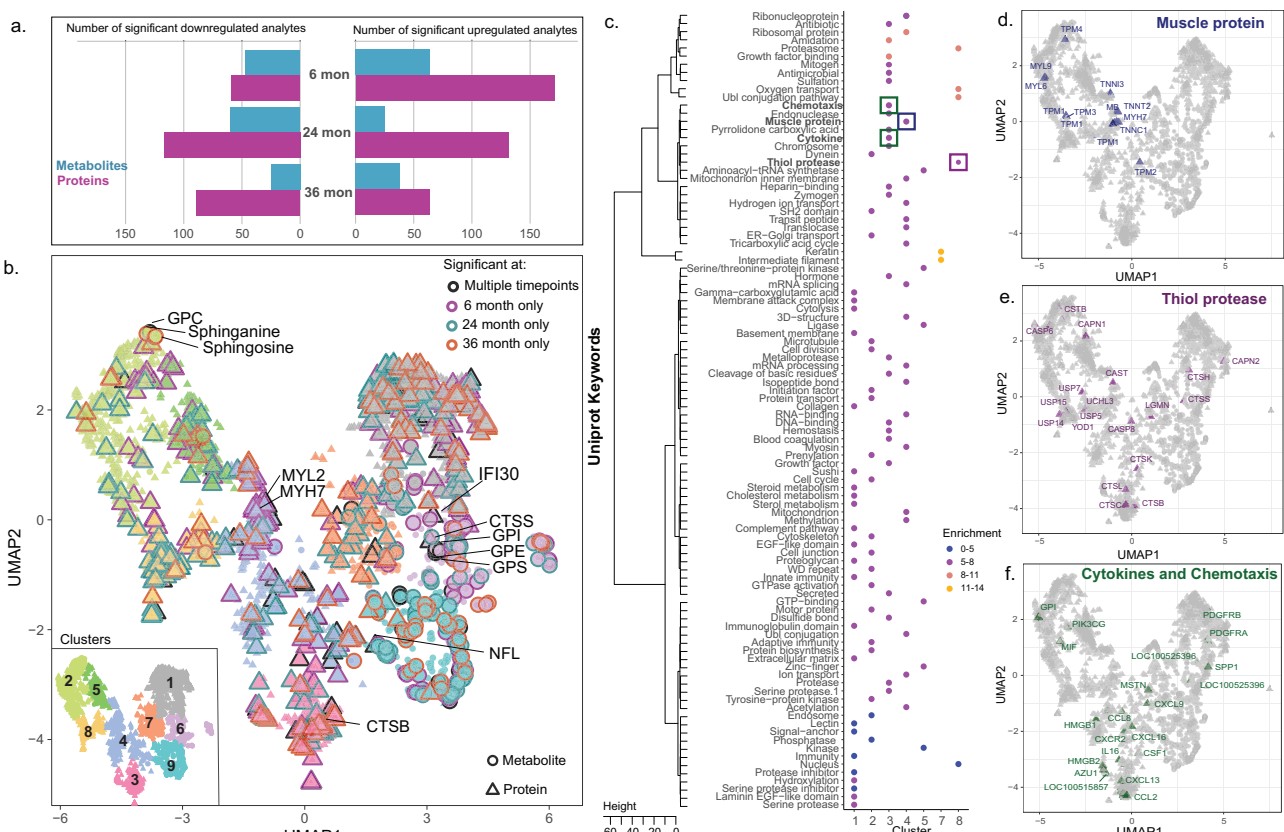

**Fig. 2 | The molecular landscape of longitudinal metabolomics and proteomics of the *CLN3* and controls over time. a** Representation of the metabolites and proteins that significantly differentiate controls and *CLN3* at 6-, 24-, and 36-months (*n* = 6, 9, and 9, respectively). **b** Uniform Manifold Approximation and Projection (UMAP) of all metabolites (circles) and proteins (triangles) from all time points that have been divided into 9 clusters using hierarchical clustering from the Euclidean distance within the UMAP. Each cluster is represented by a different color (1 = grey, 2 = light green, 3 = pink, 4 = light blue, 5 = green, 6 = light purple, 7 = orange, 8 = yellow and 9 = light teal). The features that significantly differentiate controls from *CLN3* are outlined based on if they are significant at multiple time points (black), significant at the 6-month time point only (purple), 24-month time point only (teal), or significant at the 36-time point only (orange). Metabolites and proteins of interest are labeled

with abbreviated names. **c** All proteins were mapped to their associated Uniprot Keywords. The clusters were tested for enrichment of keywords. The heatmap shows an overview of the enriched keywords in the clusters with enrichments greater than 3 and FDR corrected *p*-values less than 0.1, where the color of the circle is based on the enrichment. Enrichments of keywords connected to disease biology (shown in boxes) are represented in (**d**) muscle protein (dark blue), **e** thiol protease (dark purple) and **f** cytokine/chemotaxis (dark green). Note that the "GPI" analyte in plot 2F refers to the protein glucose 6-phosphate isomerase (GPI) rather than the gly-cerophosphoinositol (GPI) metabolite discussed elsewhere. All abbreviated protein names reflect official gene names under Human Genome Organization (HUGO). Exact *p*-values for Fig. 2 can be found in Supplementary Data 1.

mass spectrometer equipped with FAIMS Pro™ Interface (Thermo Fisher Scientific). Peptides were separated using an analytical C18 Aurora column (75 μm × 250 mm, 1.6 μm particles; IonOpticks) at a flow rate of 300 nL/min using a 80-min gradient: 1% to 6% B in 0.5 min, 6% to 23% B in 50 min, 23% to 34% B in 29 min, and 34% to 48% B in 0.50 min (*A* = FA 0.1%; *B* = 80% ACN: 0.1% FA). The mass spectrometer was operated in positive data-dependent acquisition mode, and the FAIMS Pro Interface device was set to standard resolution with the temperature of FAIMS inner and outer electrodes set to 100 °C. A three MS experiment method was set up where each experiment utilized different FAIMS compensation voltages: −45, −65, and −80 Volts, and each of the three experiments had a 1 s cycle time. A high-resolution MS1 scan in the Orbitrap (*m/z* range 350 to 1500, 60k resolution, AGC 4e5 with maximum injection time of 50 ms, RF lens 30%) was collected in top speed mode with 1-s cycle time for the survey and the MS/MS scans. For MS/MS (MS2) spectra, ions with charge state between +2 and +7 were isolated with the quadrupole mass filter using a 0.7 *m/z* isolation window, fragmented with higher-energy collisional dissociation with normalized collision energy of 35% and the resulting fragments were detected in the Orbitrap at 50k resolution, at AGC of 5e4 and maximum injection time of 86 ms. The dynamic exclusion was set to 20 s with a 10 ppm mass tolerance around the precursor. Raw proteomics data sets have been deposited to the MassIVE data repository[19].

**Proteomics data analysis.** All mass spectra files were analyzed with SpectroMine software (Biognosys, version 2.7.210226.47784) using the TMTpro 18-plex default settings. The search criteria were set as follows: full tryptic specificity was required (cleavage after lysine or arginine residues unless followed by proline), 2 missed cleavages were allowed, carbamidomethylation (C), TMTpro (K and peptide n-terminus) were set as fixed modification and oxidation (M) as a variable modification. The false identification rate was set to 1% at peptide (or PSM) and protein levels. PSM report was exported from SpectroMine to R package tools for further analysis (data available in PX).

The R package {MSstatsTMT} version 2.2.7 was used to log₂-transform the peptide intensities, impute within-TMT mixture missing values using an accelerated failure model, perform global median normalization on the peptide data (equalizing the medians across all channels and MS runs), conduct fraction aggregation, and perform protein quantification[20]. MSstatsTMT leverages a reference channel to perform local normalization, which effectively mitigates the systematic bias among different TMT mixtures. We conducted a comparison between utilizing the mean across all samples within each mixture as an artificial reference channel or using only the pooling of 48-month-old subjects as the reference channel. The results revealed that normalization based on all samples successfully minimizes between-mixture variance and eliminates the unwanted batch effect

(Supplementary Fig. 2). Consequently, we adopted this normalization approach based on all samples[21]. Subsequently, we excluded the samples from 48-month-old subjects from downstream statistical analysis due to a lack of appropriate statistical power ($n = 3$/group). Finally, abundance values for each individual protein were scaled to have a mean of zero and unit variance. Proteins missing in 50% or more of the samples were excluded from the analysis, leaving 2630 quantified proteins.

**Statistics and reproducibility**

Analytes (proteins and metabolites) that differed in abundance between the wild type and $CLN3^{\Delta ex7-8}$ samples at each time point (6-months ($n = 6$/group), 24-months ($n = 9$/group), or 36-months ($n = 9$/group)) were identified via two-tailed Student's $t$-test. Analytes with an uncorrected $p$-value $< 0.05$ were associated with genotype. The sets of proteins associated with genotype at each time point were assessed for enrichment in GO terms using Database for Annotation, Visualization, and Integrated Discovery. The sets of metabolites associated with genotype at each time point were assessed for enrichment in chemical structure types using MetaboAnalyst[22]. In both cases, Fisher's exact test was performed with Benjamini–Hochberg false discovery rate correction, and all detected proteins or metabolites were used as the background in the enrichment analyses, as applicable.

Datasets for the proteins and metabolites were normalized, scaled and combined for a final dataset. The final dataset was reduced to a 2-dimensional uniform manifold approximation and projection (UMAP) using the uwot package in R[23]. The UMAP was then clustered using hclust (stats package in R) and the distance provided by the UMAP into 9 clusters, where the number of clusters was decided by the elbow method. The proteins from each cluster were then tested for enrichment of Uniprot Keywords using the AnnoCrawler pipeline (https://github.com/DansenCode/AnnoCrawler)[24]. Significant enrichment was determined by an enrichment of a keyword in comparison to the background (the rest of the UMAP) with an FDR cutoff of 0.1.

**Multiblock sparse partial least squares discriminant analysis**. Multiblock sparse partial least squares discriminant analysis (sPLS-DA), also known as DIABLO (Data Integration Analysis for Biomarker discovery using Latent variable approaches for Omics studies), was performed on the processed proteomics and metabolomics datasets using the {mixOmics} package in R[25,26]. This technique finds a linear combination of input variables from two or more omics datasets that reduces the dimensionality of the datasets while maximizing the covariance of the reduced datasets with each other and with an outcome variable. Specifically, in the case of a centered and scaled metabolomics dataset $X^{(M)}(N \times P_1)$ and centered and scaled proteomics dataset $X^{(P)}(N \times P_2)$ with genotype labels $Y$ ($N \times G$), where $N$ is the number of samples, $P_1$ is the number of metabolites, $P_2$ is the number of proteins, and $G$ is the number of groups, for each dimension $h = 1, \ldots, H$ multiblock sPLS-DA finds the loading vectors $a_h^{(M)}$ and $a_h^{(P)}$ that maximize the value of:

$$C_{M,P} \cdot \text{cov}\left(X_h^{(M)}a_h^{(M)}, X_h^{(P)}a_h^{(P)}\right) + C_{M,Y} \cdot \text{cov}\left(X_h^{(M)}a_h^{(M)}, Y\right) + C_{P,Y} \cdot \text{cov}\left(X_h^{(P)}a_h^{(P)}, Y\right) \quad (1)$$

subject to $\| a_h^{(M)} \|_2 = \| a_h^{(P)} \|_2 = 1$, $\| a_h^{(M)} \|_1 \leq \lambda_h$, and $\| a_h^{(P)} \|_1 \leq \lambda_h$

The values of the constants $C_{M,P}$ $C_{M,Y}$, and $C_{P,Y}$ are chosen to reflect the expected degree of association between the different -omics datasets and between the -omics datasets and genotype. In this case, all constants were set to one (i.e., a fully connected design matrix was used). The value of $\lambda_h$ is chosen to constrain the number of non-zero elements in the loading vectors $a_h^{(M)}$ and $a_h^{(P)}$.

In this way, multiblock sPLS-DA selects a subset of variables (those with non-zero coefficients in $a_h^{(M)}$ and $a_h^{(P)}$) that are correlated both within and between the proteomics and metabolomics datasets and that define, for

each omics dataset $X^{(O)}$ and each dimension $h$, the component score:

$$t_h^{(0)} = X_h^{(0)}a_h^{(0)} \quad (2)$$

The process is iterative, such that the loading vector for the first dimension, $a_1^{(O)}$, is found by maximizing Eq. 1 with $X_1^{(O)} = X^{(O)}$, while the loading vector for the second dimension, $a_2^{(O)}$, is found by maximizing Eq. 1 with $X_2^{(O)} = X_1^{(O)} - t_1^{(O)}a_1^{(O)}$, and so on for however many dimensions are desired. This ensures that the first components $t_1^{(M)}$ and $t_1^{(P)}$ encompass the greatest variation in the dataset, followed by the second components $t_2^{(M)}$ and $t_2^{(P)}$, etc.

The implementation of multiblock sPLS-DA in the {mixOmics} package creates a classifier that calculates the components $t_h^{(O)}$ for a new sample, makes one classification for that sample per omics dataset in the model (based on the distance between the component scores for the new sample and the stored component scores for the training dataset points), then makes a final classification based on either majority vote, weighted vote, or averaged vote of the individual omics classifiers.

Because we desired a continuous disease score that incorporated both the proteomics and metabolomics datasets in our study, rather than the binary classifier created by default in mixOmics, we created an sPLS score by summing together the first component scores for the metabolite and protein datasets.

$$\text{sPLSscore} = t_1^{(M)} + t_1^{(P)} \quad (3)$$

We used this sPLS score to assess the combined ability of the proteins and metabolites selected in the loading vectors to distinguish $CLN3^{\Delta ex7-8}$ from wild type samples in a held-out test set.

Prior to performing multiblock sPLS-DA, the data were randomly split into a training set (70%, $n = 34$ animals) and test set (30%, $n = 14$ animals). No individual subject was represented in both the training and test sets (i.e., if a subject was sampled at multiple time points, the data for both time points were included together in either the training or test set). Multiblock sPLS-DA was performed on the training set using two dimensions ($h = 1,2$). The number of non-zero coefficients to use in the loading vectors was determined by assessing the balanced error rate of the mixOmics-generated classifier on the training dataset with fivefold cross-validation repeated 20 times. The lowest average balanced error rate ($0.004 \pm 0.010 \pm$ SD) was achieved by using one protein in the protein first component and three metabolites in the metabolite first component. However, we chose to include two proteins in the protein first component (two non-zero coefficients in $a_1^{(P)}$) and two metabolites in the metabolite first component (two non-zero coefficients in $a_1^{(M)}$) because this also yielded a very low average balanced error rate ($0.026 \pm 0.023$) and presented the benefit of drawing two analytes from each of our omics datasets. A single analyte was used for each second component (one non-zero coefficient each in $a_2^{(P)}$ and $a_2^{(M)}$) because the number of analytes selected for the second components had little impact on classification accuracy, and because the second components tended to correlate with age rather than genotype.

For the proteins, the first component was initially a linear combination of the normalized abundance values for CTSS and ADAMTSL4. However, due to lack of availability of commercial kits for testing ADAMTSL4 levels in patient samples, we removed ADAMTSL4 from consideration and repeated the multiblock sPLS-DA workflow.

**Causality analysis**. Analytes (proteins and metabolites) that differed in abundance between the wild type and $CLN3^{\Delta ex7-8}$ samples at each time point (36-months) were identified via two-tailed Student's $t$-test. Median imputation was performed for missing values. 152 proteins and 63 metabolites were used in the network analysis. The $\log_2$ fold change values for analytes of individual $CLN3^{\Delta ex7-8}$ animals as compared to average wild type values were discretized using Cyni Toolbox Equal Width/ Frequency Discretization with 20 intervals[27,28]. The Bayesian–Hill Climbing inference algorithm with Bayesian Dirichlet Equivalent (BDe)

metric was used with the max number of parents as 3, reverse edges; only nodes with edges were outputted. The hill-climbing algorithm with a local search built the causal Bayesian network[29]. The algorithm began with only nodes and iterates through the possibilities of adding, removing, or reversing an edge until the highest probability as measured with the BDe metric score is found[28]. This process is repeated until no improvement is found. The BDe metric does not require prior expert knowledge and is based on likelihood equivalence[30]. Networks were organized using the Perfuse Force Directed layout; scores from the Bayesian Hill Climbing algorithm were used as weight and were evaluated by the Heuristic function (min edge weight:0, max weight:1, default:0.5, num of iterations:100, spring coefficient: 1E-4, spring length:75, node mass:5, force deterministic layouts). The displayed edge weights correspond the BDe metric score; higher scores resulted in thicker edges. Subnetworks were created by selecting edges with scores greater than 0.001. The connected nodes and those nodes neighbors (directed: incoming) were selected.

## Reporting summary
Further information on research design is available in the Nature Portfolio Reporting Summary linked to this article.

## Results
To investigate longitudinal blood-based CLN3 disease signatures, we performed deep multi-omics profiling to quantify relative blood serum concentrations of 769 metabolites and 2634 proteins in samples from male and female wild type and homozygous $CLN3^{\Delta ex7-8}$ Yucatan minipigs. An average of 2398, 2438, 2431 proteins and 738, 747, 742 metabolites were detected in 6-, 24-, and 36-month WT animals, respectively, and 2430, 2437, 2439 proteins and 729, 744, 741 metabolites were detected in 6-, 24-, and 36-month $CLN3^{\Delta ex7-8}$ animals, respectively. We selected 6-, 24-, and 36-month time points to represent early-stage disease prior to any appreciable visual or motor defects, mid-stage/symptomatic, and late-stage disease, respectively (Fig. 1). Of the 3403 analytes quantified in the dataset, 230 proteins (171 upregulated; 59 downregulated) and 111 (64 upregulated; 47 downregulated) metabolites are associated with genotype at the 6-month time point, suggesting early systemic changes (Supplementary Figs. 3 and 4). At 24-months, 249 proteins (132 upregulated; 117 downregulated) and 85 metabolites (25 upregulated; 60 downregulated) are associated with genotype, while at the 36-month timepoint 153 proteins (64 upregulated; 89 downregulated) and 63 metabolites (38 upregulated; 25 downregulated) are associated with genotype (Fig. 2a). To discern large-scale relationships among the combined metabolomics and proteomics datasets, we employ the UMAP algorithm to reduce the data into a 2-dimentional space (Fig. 2b), and Bayesian Network Analyses to build putative networks (Supplementary Figs. 5–7). The UMAP algorithm is used because it is effective in visualizing clustering patterns in high-dimensional data, therefore, the distance between the features is relative to the way the analytes co-vary in the dataset and associate in the 2-dimentional space. The features are colored by cluster and have bold outlines if they are found to be significant at one or more time points. The features in the UMAP are separated into 9 distinct clusters, each enriched for certain Uniprot Keywords (Fig. 2c). Shown are enrichments greater than 3 with an FDR corrected $p$-value less than 0.05, with the full list in Supplementary Data 1. Specific Uniprot keywords with a clear rationale connecting them to disease biology (Muscle protein, Thiol protease, Cytokine/Chemotaxis) are selected to show the enrichment throughout the UMAP of selected relevant biological features (Fig. 2d–f). Consistent with previous studies, we identify a group of structurally similar glycerophosphodiesters (GPI, GPE, GPS; Fig. 2b) among the most significantly and consistently upregulated species across all time points, while a group of phospholipids sharing a docosahexaenoyl

group at the sn2 position are among the most significantly and consistently downregulated analytes[8].

In addition to these previously identified analytes, deep nanoparticle-based proteomics workflow (Fig. 3a) combined with isobaric labeling and 18×multiplexing (TMT) quantified over 2600 proteins, several of which are strongly associated with Batten disease genotype (Fig. 3b–g). The top two differentially expressed proteins across all time points are Cathepsin S (CTSS) and Cathepsin B (CTSB), lysosomal cysteine proteases that both demonstrate remarkably stable longitudinal patterns of elevation (Fig. 3b, c). Although not significant at all time points, gamma-interferon inducible lysosomal thiol reductase IFI30 (IFI30), Myosin regulatory light chain 2 (MYL2), and Myosin-7 are also elevated in CLN3 mutant samples (Fig. 3d–f). As previous studies have shown, NFL is able to significantly differentiate controls and CLN3 disease but only at 36-months (Fig. 3g). Other established markers of neurodegenerative disease (i.e., TAU, GFAP, UCHL1) are not significantly altered. Using the streamlined nanoparticle-based proteomics workflow provides several advantages. In contrast to targeted affinity-based approaches that require species-specific affinity probes, the nanoparticle workflow is directly amenable to non-human model organisms. Moreover, direct compatibility with multiplexing at the peptide level enhanced throughput and data robustness. Together, this enables one of the most comprehensive—omics profiles of a CLN3 model, and the deepest omics-profile of porcine serum to date, quantifying putative disease biomarkers with a depth of coverage not typically achievable with standard serum proteomics workflows[14,15,31] (Fig. 3h and Supplementary Table 1).

To elucidate the functional relevance of the differentially expressed proteins, we probed for enriched functional sets of proteins and pathways within the subsets of proteins associated with genotype at each time point. Three 'cellular component' signatures are flagged at all time points: cytoplasm, proteasome, and secreted (Supplementary Fig. 8). Interestingly, the signature "lysosome" is shared only between the 24- and 36-month groups suggesting that blood signatures of lysosomal dysfunction increase over time, perhaps reflecting progressive dysfunction in this organelle. When categorized by "biological process," the related term "chemotaxis" is shared among all groups, suggesting alterations in immune homeostasis. (Supplementary Fig. 9). Unique to the 36-month time point, we find a large network of proteins and metabolites involved in sphingolipid metabolism. Characterized by a common eighteen-carbon amino-alcohol backbone, sphingolipids are crucial to a vast number of biological processes including cell signaling and membrane structure and have previously been linked to CLN3 disease, establishing another putative functional link between biomarker signatures and pathogenicity[32–34].

Kyoto Encyclopedia of Genes and Genomes (KEGG) pathway analysis confirm the enrichment for species involved in sphingolipid metabolism at 36 months. We detect the presence of 21 analytes with known functions in sphingolipid metabolism and related pathways, 11 of which are differentially abundant in CLN3 serum (Fig. 4 and Supplementary Fig. 10). Notably, all the differentially abundant metabolites are elevated in CLN3 serum, suggesting alterations in sphingolipid synthesis, degradation, or defective transport of intermediates (Fig. 4 and Supplementary Fig. 10a–f). Many proteins contributing to sphingolipid metabolism are similarly elevated, suggestive of a compensatory mechanism triggered by the accumulation of sphingolipids and resulting increased lysosomal burden (Supplementary Fig. 10g–l). Given the complex nature of this pathway, further investigation is needed to clarify the impact of CLN3 disease on sphingolipid metabolism, or vice versa.

Here, we investigate biomarkers manifesting early in disease progression, prior to the onset of symptoms which could have utility for pre-symptomatic clinical diagnosis, early evaluation of interventions, and providing insights into upstream disease processes (Supplementary Fig. 11). Of the 230 genotype-associated protein targets at 6-months, Integrin Beta-2 (ITGB2) is among the most significantly downregulated protein, although ITGB2 serum levels return towards WT levels at later time points, remaining unchanged at 24 and 36 months (Supplementary Fig. 11a). Conversely, Calpastatin (CAST) and Myosin Light Chain 3 (MYL3)—cardiac-enriched proteins—are

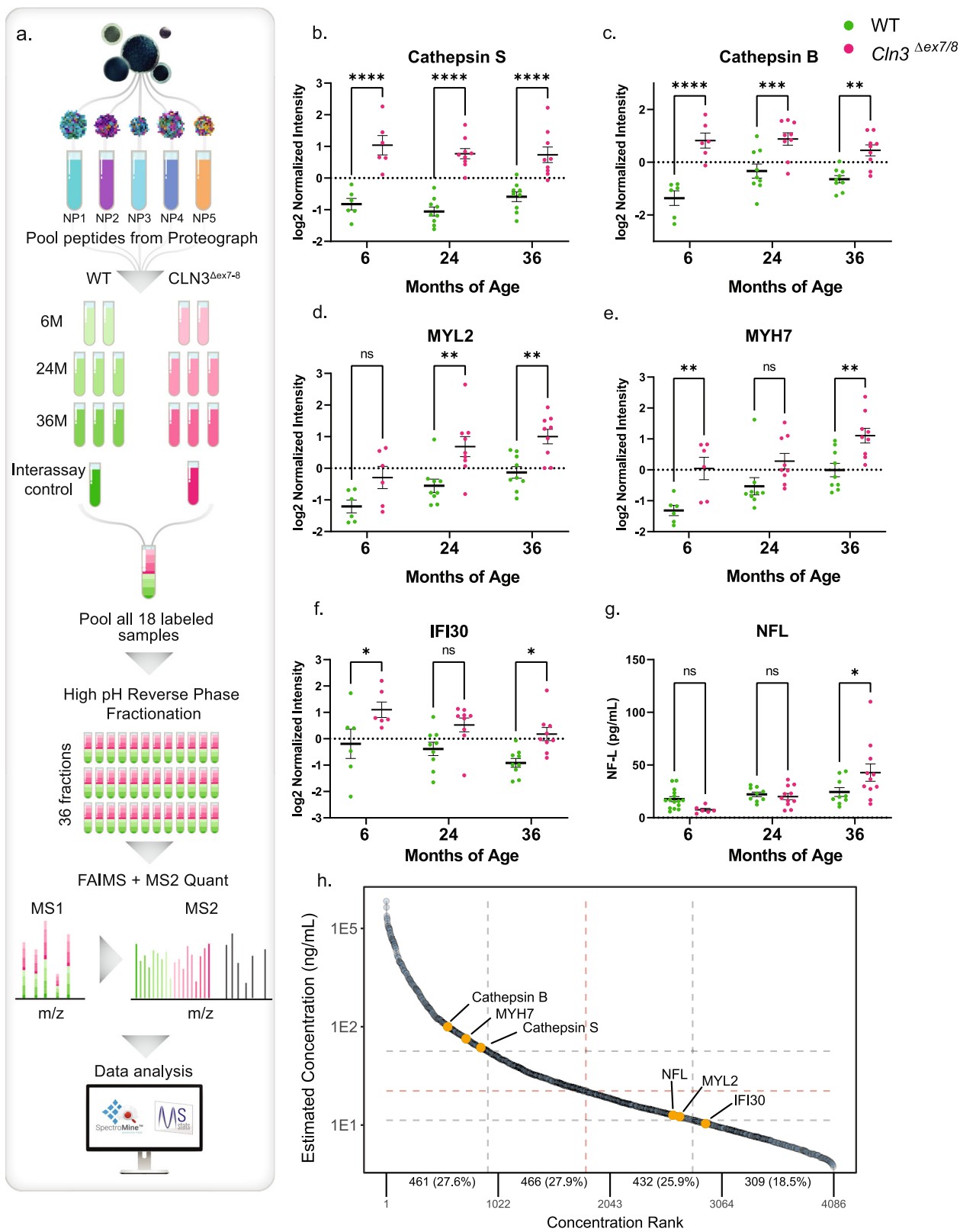

significantly elevated in the *CLN3* cohort at 6 months when compared to wild type, with smaller elevations at later time points (Supplementary Fig. 11b, c). Glucuronide of $C_{12}H_2OO_3$ and 3-aminoisobutyrate are downregulated at 6 months with a rebound at later time points (Supplementary Fig. 11d, e). Interestingly, maleate—showing the largest fold change of any upregulated metabolite species at 6 months – stabilizes to near identical levels in *CLN3*$^{\Delta ex7-8}$

and wild type minipig serum at 24- and 36-months (Supplementary Fig. 11f). Together these data demonstrate complex pre-symptomatic molecular changes at the protein, lipid and metabolite levels throughout disease progression that could serve to assess therapeutic efficacy.

Given the immense scale and complexity of the multi-omics dataset, we seek to condense these data into a disease score model that

**Fig. 3 | Proteograph workflow with TMT 18-plex mass spectrometry method and resulting protein biomarker regulations. a** Serum samples were first processed by SP100 Automation instrument with ProteographTM Assay kit included in Proteograph Product Suite (Seer, Inc.) using five distinctly functionalized nanoparticles (NPs). Tryptic peptides from 5 NPs were then pooled into one single sample for TMT labeling. A total of 54 pooled samples were allocated into three 18-plex TMT mixtures, each of which contained two 6-month samples, three 24-month samples, three 36-month samples and one interassay control sample from WT and $CLN3^{\Delta ex7-8}$, respectively. Each TMT mixture was followed by high pH reverse phase fractionation and LC-MS/MS analysis, comprised of a Proxeon EASY nanoLC system coupled to an Orbitrap Fusion Lumos MS equipped with FAIMS Pro Interface (Thermo Fisher Scientific). The raw spectra data were finally processed by SpectroMine (Biognosys) and R/Bioconductor package MSstatsTMT to generate statistical analysis results. **b–f** Selected biomarker candidates, lysosomal proteins cathepsin S

(CTSS), cathepsin (CTSB), and interferon gamma-inducible lysosomal thiol reductase (IFI30) and muscular proteins myosin light chain 2 (MYL2) and myosin heavy chain 7 (MHY7) are significantly elevated at multiple time points in $CLN3^{\Delta ex7-8}$ minipig serum when compared to age-matched healthy control animals, **g** while neurofilament light chain (NFL) blood-serum levels are significantly elevated at 36 months only. Two-way ANOVA with Šidák correction for multiple comparisons, 95% confidence interval, $^*p < 0.05$, $^{**}p < 0.01$, $^{***}p < 0.001$, $^{****}p < 0.0001$, $n = 6; 9; 9$ animals respectively, mean +/- SEM. Exact $p$-values can be located in Supplementary Data 10. **h** The 2634 identified proteins were mapped to the Human Plasma Proteome Project (HPPP) protein database, revealing their detection throughout the entire concentration range of the database. Notably, the proteins, MYL2, and IFI30 were found to be within the low abundance range. CTSS, CTSB, IFI30, MYL2, and MHY7 detected via TMT 18-plex Mass Spectrometry. NFL quantified via Simoa® targeted Neurology 4-PlexA assay.

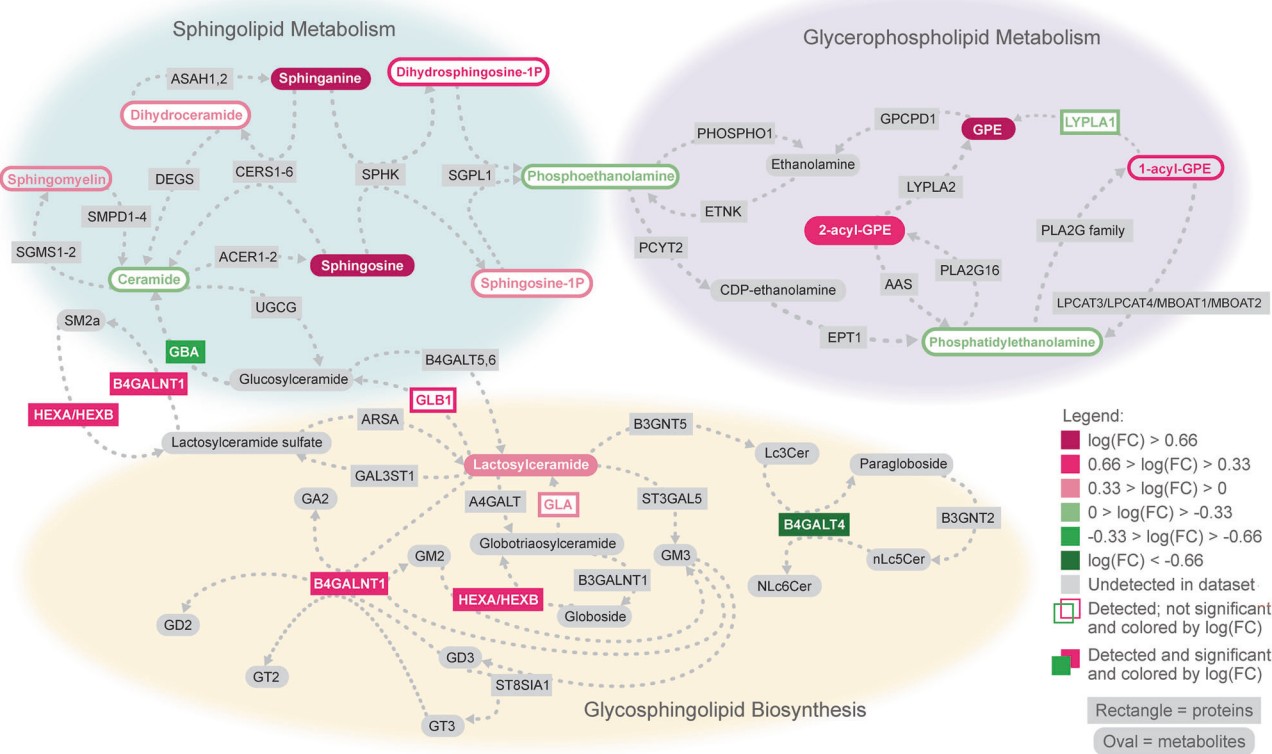

**Fig. 4 | Sphingolipid alterations at 36 months in CLN3 disease.** Network of proteins and metabolites involved in sphingolipid metabolism and related pathways that were differentially abundant in serum of 36-month-old subjects ($n = 9$). Network connections were derived from the Kyoto Encyclopedia of Genes and Genomes (KEGG) pathways for sphingolipid metabolism, glycerophospholipid metabolism, and glycosphingolipid biosynthesis. Serum levels of beta-1-4-galactosyltransferase 4 (B4GALT4) and glucocerebrosidase (GBA) were significantly lower in $CLN3^{\Delta ex7-8}$ pigs than in WT pigs, while serum levels were significantly higher for beta-1,4-N-acetyl-galactosaminyltransferase 1 (B4GALNT1), hexosaminidase subunit alpha (HEXA), hexosaminidase subunit beta (HEXB), lactosylceramide, sphinganine, sphingosine, glycerophosphoethanolamine, sphingomyelin, and 1-acyl-GPE. Analytes not significantly different between genotypes are outlined according to the magnitude and direction of their log fold change, while analytes involved in the

pathway that were not detected in the dataset are presented in gray. Many of the metabolites in the KEGG pathways are general metabolite categories rather than specific chemical species (e.g., ceramide, 1-acyl-GPE). Multiple analytes were detected in the dataset that fall under the categories of 1-acyl-GPE ($n = 5$), ceramide ($n = 6$), phosphatidylethanolamine ($n = 10$), and sphingomyelin ($n = 28$). Not all detected analytes within these categories behaved the same way; the color of the category in the figure is based on the overall trend. Other categories had only one representative detected in the dataset, including 2-acyl-GPE (2-stearoyl-GPE (18:0)), dihydroceramide (N-palmitoyl-sphinganine (d18:0/16:0)), and lactosylceramide (lactosyl-N-palmitoyl-sphingosine (d18:1/16:0)). All abbreviated protein names reflect official gene names under Human Genome Organization (HUGO). Exact $p$-values can be found in Supplemental Data 3, and 4.

could reflect overall disease signatures derived from a minimal set of input variables. In this study, similar to the utilization of multi-domain responder indices that have gained popularity in clinical trials, multiblock sPLS-DA is employed to reduce the proteomics and metabolomics training datasets into two components each. This allows us to build a four-component score consisting of both proteins and metabolites to serve as a proof of concept for a simpler biomarker

score obtainable with targeted assays. A tool such as this could have enhanced feasibility for disease screening as a clinical endpoint[25,26]. For the proteins detectable with the nanoparticle-based workflow, the first component comprises CTSS and CTSB (Fig. 5A). For the metabolites, the first component is a linear combination of the normalized abundance values for glycerophosphoinositol (GPI) and glycerophosphoethanolamine (GPE) (Fig. 5B). For both datasets, the first

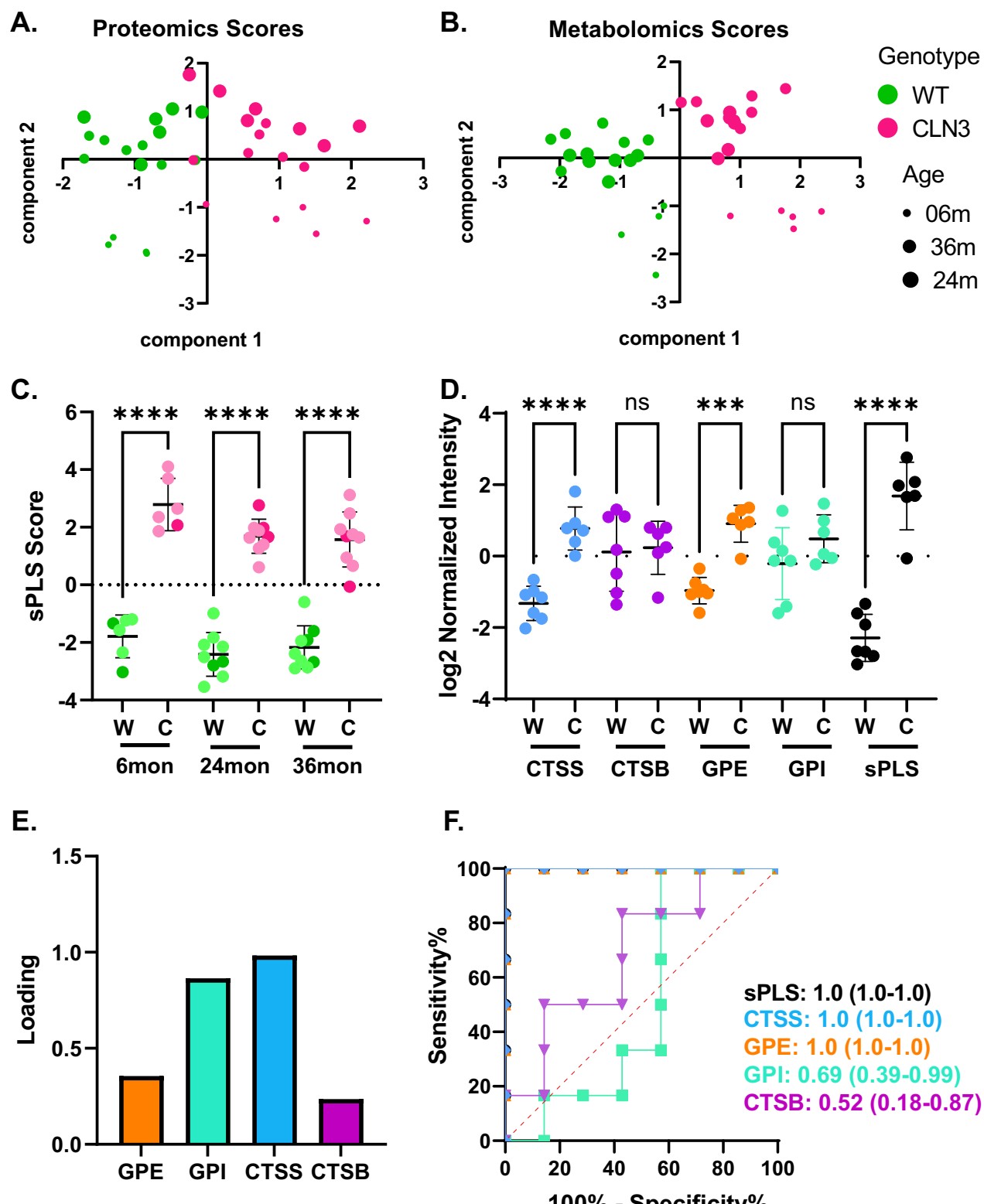

component separates the samples by genotype, while the second component separates the samples by (Fig. 5A, B).

The predictive accuracy of the model is assessed with a held-out set of test samples, where sPLS scores are significantly different between $CLN3^{\Delta ex7-8}$ and wild type samples (two-tailed $t$-test, $p < 0.0001$) from all three time points (Eqs. 1–3) (Fig. 5C, D). When all time points are analyzed in aggregate, the sPLS score serves as a perfect

classifier for the test set (ROC AUC = 1.0, Supplementary Data 2) and separates the wild type and $CLN3^{\Delta ex7-8}$ samples in the test set better than any individual analyte used to construct the sPLS score (Fig. 5D). The weight for each of the selected analytes in the model can be seen in the loadings plot (Fig. 5E). The ROC curves (Fig. 5F) validate these findings by showing perfect separation for the sPLS score. (ROC AUC = 1.0).

**Fig. 5 | Multiblock sPLS-DA loadings highlighting protein biomarkers for *CLN3*.**
**A**, **B** Training data projected onto the two proteomics components and two meta-bolomics components identified using multiblock sPLS-DA (Sparse Partial Least Squares Discriminant Analysis). For both omics datasets, the first component separated the datapoints by genotype, and the second component separated the datapoints by age ($n = 6$, 9, and 9, respectively). **C** sPLS scores calculated for the entire dataset, with test set points shown in dark green/magenta. At each time point, sPLS scores were significantly different between WT and $CLN3^{\Delta ex7-8}$ subjects in the dataset as a whole. Scores were also significantly different between WT and $CLN3^{\Delta ex7-8}$ subjects in the test set at all time points combined ($p < 0.0001$, Student's two-tailed *t*-test), demonstrating the robustness of the sPLS score as a tool for differentiating $CLN3^{\Delta ex7-8}$ from WT samples. Exact *p*-values can be located in Supplementary Data 10. **D** Comparison of sPLS score and log2 normalized intensity values of the analytes that make up the sPLS score for separating data in the test set.

The sPLS score better separates the test set WT from $CLN3^{\Delta ex7-8}$ samples compared to any individual analyte (bars represent standard deviation; asterisks represent *p*-values from two-way ANOVA with Tukey's post-hoc test). Exact *p*-values can be located in Supplementary Data 10. **E** Loadings of the analytes in the sPLS score created by combining the formulas for the protein first component and metabolite first component. The formula for the protein first component relied on cathepsin S (CTSS) and cathepsin B (CTSB), while the formula for the metabolite first component relied on glycerophosphoinositol (GPI) and glycerophosphoethanolamine (GPE). The sPLS score formula places the greatest emphasis on CTSS, followed by GPI, then GPE, and finally CTSB. **F** Receiver operating characteristic curves for each of the analytes included in the sPLS score as classifiers for identifying $CLN3^{\Delta ex7-8}$ samples in the test set. The area under the ROC curve is presented for each analyte along with its 95% confidence interval (computed with Wilson/Brown method). CTSS and GPE both serve as perfect classifiers for the test set.

## Discussion

The absence of functional CLN3 gives rise to an intricate disease state commonly referred to as Batten disease. Cells lacking functional CLN3 protein exhibit a wide range of abnormalities including trafficking deficits within the secretory pathway, changes in lipid composition, impaired autophagy, and altered lysosomal composition and function[34–37]. At the tissue level, the central nervous system is progressively impaired by exten-sive neuroinflammation[38], while there is some evidence of progressive dysfunction in the heart, skeletal muscles, and immune cell populations in the periphery[39]. Despite comprehensive characterization of disease signs and symptoms, there is little consensus around the primary, secondary, or tertiary etiologies emerging from CLN3 dysfunction. Furthermore, there is a notable scarcity of information on clinically measurable biomarkers that could be used to monitor disease progression or therapeutic efficacy.

Our study addresses these knowledge gaps with longitudinal deep multi-omics profiling using blood serum from a large animal model of CLN3 disease, the $CLN3^{\Delta ex7/8}$ minipig. The $CLN3^{\Delta ex7/8}$ minipig exhibits progressive disease pathology and behavioral abnormalities, more closely resembling human disease as compared to currently available small animal models such as the $Cln3^{\Delta ex7/8}$ mouse. Additionally, the use of $CLN3^{\Delta ex7/8}$ minipig in biomarker studies offers a controlled, isogenic, and well-powered approach compared to the challenges of studying rare samples from indi-viduals with CLN3 disease with wide genetic variation. A particular chal-lenge with animal models is the extraordinarily large dynamic range of protein concentrations in blood and lack of appropriate affinity probes to either deplete or enrich proteins to facilitate comprehensive proteome capture at scale. Our study demonstrates the utility of a nanoparticle-based proteomics workflow that mitigates the dynamic range challenges in a species-agnostic way to facilitate unbiased and deep serum proteomics. In combination with global metabolomic and lipidomic profiling, we quantify over 3400 analytes longitudinally (2634 proteins; 769 metabolites and lipids), significantly surpassing previous porcine multi-omic studies, which were hindered by the limitations of human- or mouse-specific depletion kits[40,41]. The combination of the Proteograph workflow with TMTpro 18-plex reagents has enables the simultaneous analysis of 18 samples, increasing the throughput and accuracy of protein quantification with enhanced data completeness[42]. Further, our workflow for multiplexed proteomics data acquisition using MS2 methods on an Orbitrap Tribrid MS instrument utilizing the FAIMS Pro interface has shown to enhance quantification accuracy[43]. The vast majority of dysregulated analytes are, to the best of our knowledge, novel in comparison to those detected in prior studies using mouse or human samples with some limited overlap.

With this unique approach, we are able to identify biomarker sig-natures that offer insights into the cellular dysfunction and tissue pathology associated with the loss of functional CLN3, as well as a more detailed understanding of the disease timeline. The early emergence and steady maintenance of elevated glycerophosphodiesters suggests a close association with CLN3 function, while later in the disease course a second signature comprised of sphingolipid metabolism proteins and metabolites emerges

(36-months only). A persistent lysosomal signature is also present throughout disease progression, likely due to increased lysosomal mass (from storage material accumulation and upregulation of lysosomal bio-genesis) either exocytosed or released into circulation by progressive inflammation and tissue damage.

Although the role of glycerophophodiesters is not completely under-stood in Batten disease, elevated levels have been observed early in the course of the disease and are maintained at steady levels, suggesting a close asso-ciation with CLN3 function. One likely possibility is that glycerophos-phodiesters are an early lysosomal storage substrate, leading to subsequent secondary storage of additional proteins, lipids, and metabolites. If this is the case, targeting proteins at the upstream metabolic pathways related to gly-cerophosphodiester metabolism may have therapeutic potential. Later in disease progression, effective treatments may need to target additional facets of disease such as immune activation, aberrant sphingolipid metabolism, and downstream lysosomal pathology. Importantly, our data indicates that some of the molecular patterns are temporally restricted, suggesting that an ideal therapeutic strategy could potentially be tailored to an individual's specific biomarker signature of disease progression.

We discovered several proteins, many of which would be difficult or impossible to detect with conventional unbiased proteomics workflows at scale (Supplementary Table 1), exhibiting differential expression between CLN3 disease and control samples[15]. Among them, lysosome-associated CTSS and CTSB are consistently upregulated at all time points which is consistent with the central role of lysosomal dysfunction in the disease pathology. Another lysosomal protein, IFI30, exhibits similar elevations, although differences did not reach significance at the 6-month time point. Why these lysosomal proteases are enriched in a peripheral biofluid remains unclear. One likely possibility is that CTSS, CTSB, and IFI30 are enriched in disease-affected cells, possibly as a consequence of increased lysosomal mass or as a compensatory mechanism for clearing storage material. Either of these would be an expected downstream consequence of Transcription Factor EB (TFEB) activation as has been observed in other lysosomal storage disorders (LSDs) CLN3 disease[44]. The proteins could then be released into circulation upon lysosomal exocytosis or cell death. It is unclear as to why some lysosomal enzymes are affected more than others. This could be an artefact of amenability to our mass spec workflow or a genuine biological phenomenon that require further investigation.

The cardiac-enriched proteins MYL2 and MYH7 show similar pat-terns of elevation, reaching significance at most time points. It is tempting to speculate that these markers could reflect a cardiac defect in this model—a hypothesis that warrants further investigation. Cardiac phenotypes (left ventricular hypertrophy, bradycardia, storage material) have been described in CLN3 affected individuals, but a lack of strong phenotypes in mouse models has thus far precluded exhaustive preclinical study of this facet of disease[45].

These protein-based biomarkers could be valuable additions to the repertoire of existing markers for CLN3 disease. Associated with late-stage disease (i.e., neurodegeneration), one existing marker, NFL, offers the

**Article**

benefit of being closely linked to the central neuronal pathology of the disease, but exhibits only mild elevations in affected individuals and animal models, as reflected in our data here. The glycerophosphodiesters are appealing biomarker candidates due to their early and steady elevation, but little is known about their relationship with disease etiology. In combination, however, CTSS, CTSB, GPI, and GPE offer a powerful summary of disease state from a limited set of variables, as evidenced by our sPLS score model. Further development of a CLN3 disease biomarker panel could also incorporate markers of neuronal damage, such as NFL, to build an even more comprehensive picture of an individual's disease status, particularly at later-stage disease.

Our data also provides insights into the pathogenic timeline of Batten disease. While CTSS, CTSB, GPI, and GPE are elevated at all time points, other signatures develop only at later stages of disease progression. For example, MYL2 and NFL are significantly upregulated only in the later time points, suggesting that cardiac and neuronal damage may take substantial time to initiate. Additionally, sphingolipid metabolism is only enriched at the 36-month time point, suggesting that sphingolipid accumulation takes longer to develop and may contribute to disease pathology in a more tissue-specific manner before it is a useful biomarker in blood. Previous studies have similarly found sphingolipid metabolism to be dysregulated in various models of CLN3 disease, although exactly how this process contributes to disease remains unclear[46,47]. Regardless of the mechanism, treating these secondary or tertiary disease manifestations with targeted therapies may be beneficial alongside disease-modifying therapies that compensate for the loss of CLN3.

Collectively, this work demonstrates how deep multi-omics profiling can be employed to detect disease-specific biomarkers related to cellular dysfunction and tissue pathology associated with a complex degenerative disease. Innovative techniques to mitigate interference of highly abundant proteins in pig serum are crucial in enabling comprehensive proteomics. The power of this technique is evidenced not only by the identification of proteins and metabolites as biomarker candidates but also by the insights into the timeline and order of disease progression. This study's findings demonstrate the potential of deep multi-omics profiling for uncovering disease-specific biomarkers, which can provide valuable insights for understanding disease and facilitating the identification of potential drug targets, thus offering valuable insights for therapeutic interventions.

## Data availability

Metabolomics/lipidomics and proteomics data that support this study were deposited into the MetaboLights and MassIVE repositories, respectively. Metabolomics/lipidomics data are available under accession number MTBLS1104 and at the following URL: https://www.ebi.ac.uk/metabolights/MTBLS1107. Proteomics data are available under accession number PXD044146 and at the following URL: https://massive.ucsd.edu/ProteoSAFe/dataset.jsp?task=c0856cd9532f47cfa86554f54ce91870. All data generated or analyzed during this study are included in this published article (and its supplementary information files). The custom code used to generate and analyze data in this study is available from the Zenodo repository at https://doi.org/10.5281/zenodo.3939280[24]. The source data for Fig. 2 is in Supplementary Data 1. The source data for Fig. 3 and Supplementary Fig. 10, and 11 is in Supplementary Data 5, and 6. The source data for Fig. 4 and Supplementary Figs. 4, and 5 is in Supplementary Data 3, and 4. The source Data for Fig. 5 is in Supplementary Data 11. Source data for Supplementary Fig. 1 is in Supplementary Data 6. The source Data for Supplementary Figs. 5, 6, and 7 is in Supplementary Data 7, 8, and 9, respectively. All other data are available from the corresponding author upon reasonable request.

## Code availability

The custom code used to generate and analyze data in this study is available from the Zenodo repository at https://doi.org/10.5281/zenodo.3939280[24].

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

## Acknowledgements

Funding for this work was provided by the ForeBatten Foundation as a grant to Sanford Research.

## Author contributions

The project was conceived by J.J.B. and J.M.W. J.J.B. designed the experiments. Experiments were performed by M.J.R., B.L., C.N., T.H., A.R.C., and R.D. The manuscript was prepared by M.J.R, B.L, C.N, T.H, A.R.C, K.M, D.H, T.B.J, and J.J.B. M.J.R, B.L, C.N, T.H, A.R.C, K.M, D.H, T.B.J, V.J.S, J.M.W, and J.J.B reviewed and edited the manuscript.

## Competing interests

The authors declare the following competing interests: J.M.W., J.J.B., T.B.J., and C.N. are employed by Amicus Therapeutics, Inc. and hold equity in the company in the form of stock-based compensation. T.H. has a financial interest in Seer. The remaining authors declare no competing financial interests.
