## [Transparent Peer Review file · Communications Medicine]

Longitudinal Deep Multi-Omics Profiling in a CLN3 Δ ex7/8 Minipig Model as a Pipeline to Identify Novel Biomarker Signatures

Corresponding Author: Dr Jon Brudvig

Version 0:

Reviewer comments:

Reviewer #1

(Remarks to the Author)

The authors have presented a valuable contribution to the field by conducting a comprehensive multi-omics analysis of a CLN3 Δ ex7/8 minipig model. The methodological approach is sound, and the generated data provide a rich resource for future investigations. To enhance the impact and clarity of the manuscript, some aspects should be addressed:

Major Considerations:

Title: While the current title is informative, it might be slightly overstated. Consider revising it to better reflect the scope and depth of the findings. To keep it in its current form human validations are mandatory.

Comparison with Existing Literature: The authors should explicitly address how their findings align with or diverge from previous studies on Batten disease, particularly those employing omics approaches. A comparative figure illustrating the overlap and differences between their results and published data would be beneficial. For example, compare their findings to the transcriptomic data in PMID: 38853929, PMID: 32592935, PMID: 30771446, and the metabolomic data in PMID: 38195117, PMID: 36212622 (from Brudvig's previous work), PMID: 20485751, PMID: 16239221.

Causality Analysis: While the multi-block analysis was useful for reducing the dimensionality of the data, exploring causal relationships between the identified biomarkers is crucial for understanding disease mechanisms. Employing Bayesian or other probabilistic networks could provide valuable insights in this regard. Please build them. Some published models include PMID: 39288230, PMID: 39251098, PMID: 38579163.

The sex of the animals was not mentioned. Are these findings valid for male and female Batten disease pigs?

Minor Points:

IACUC Number: Please ensure that the IACUC number is included in the manuscript.

The increase in lysosomal enzyme activities observed in the study is likely due to TFEB-mediated transcription. Discussing this aspect and addressing why only certain cathepsins were affected, while sphingolipid metabolism enzymes remained relatively unchanged at late stages, would provide a more complete understanding of the findings. Refer to PMID: 19556463 for more information on TFEB-mediated transcription.

By addressing these points, the authors can significantly enhance the quality and impact of their manuscript, making it a valuable contribution to the field of Batten disease research.

Reviewer #2

(Remarks to the Author)

This study describes a detailed metabolomic and proteomic analysis of plasma from a pig model of juvenile neuronal ceroid lipofuscinosis. This appears to be a detailed and useful study but lack of access to the primary data is a major problem.

Major comments.

It is standard in the biological mass spectrometry field to make all raw, database search and result files accessible to reviewers during manuscript submission and publicly accessible after publication. While the data analysis in this submission appears sound, the ability to review the primary data, in particular raw and normalized TMT reporter intensities for proteomic analysis and label-free quantitation for metabolomics, is absolutely needed for full evaluation. This problem would be easily corrected by data submission to a public repository, e.g. MassIVE and providing reviewer access.

The study initially comprised 4 timepoints (6, 24, 36 and 48-months) but the 48-month timepoint was dropped from the study "due to limited biological variance". This merits further explanation especially as the 48-month time-point, which presumably reflects late-stage disease, could be highly informative as changes in biomarkers that may be missed at earlier ages might be expected to be more pronounced as disease progresses.

In the abstract, it is mentioned that 769 metabolites and 2,634 proteins were quantified as an aggregate of all samples but before depth of coverage can be properly assessed, it would be important to know the average number of each per sample. Along similar lines, on P16, there is discussion attributing the depth of proteome coverage to be due to the nanoparticle digestion approach but it is not possible to determine what degree of improvement comes from this method versus the two-dimensional liquid chromatography. Additional comparison studies would really be needed to justify such conclusions. Also, cathepsins S and B are readily detectable in plasma using standard proteomics workflow e.g. data-independent acquisition.

Minor comments.

P3. I agree with the authors that additional biomarker in JNCL may be valuable but the comments downplaying NFL because of "highly-variable elevations" probably needs further explanation to justify.

Fig 3. Is NFL measured by immunoassay or mass spectrometry? This should be clarified in the legend and if former, comments re. detection in low abundance range by mass spectrometry are not relevant.

Reviewer #3

(Remarks to the Author)

This manuscript from Rechtigel and colleagues under the guidance of Brudvig is a tour de force in terms of new data generation and handling for the field of lysosomal storage disorders. With a focus on CLN3, this could represent the beginnings of a more standardised pipeline for the generation of a molecular fingerprint for identifying, predicting and monitoring neurodegenerative disease progression in general. Both for tracking normal disease course and potentially in response to therapeutic intervention.

This also crucially frames the important role that larger animal model systems can play in discovery driven science with a view to support and play NDA enabling translational studies. Lysosomal storage disorders are a significant proportion of the "effective" therapies for neurodegenerative conditions. The availability of Brineuria for cIn2 from Biomarin (via a canine model) and the permissions for gene therapy for CLN5 built on livestock models for scale up, efficacy and tox simultaneously, places the use of livestock in a more prominent role in the therapeutic development pipeline (especially as an alternative to NHP tox studies).

I believe that the depth of analysis here and the approaches provided will be of significant interest for researchers, clinicians and industrial stakeholders. but I have some queries and suggestions which I would like to see in order to clarify some of the methodology and processes presented here:

Firstly, the title could/should be changed. It is quite long and maybe fails to convey that this work makes use of novel resources like the CLN3 porcine model, AND that the approach could be replicated for any other neurodegenerative condition where such a model is available. I suggest shortening to something like:

"Longitudinal Deep Multi-Omics Profiling of CLN3 Δ ex7/8 Minipig Model as a pipeline to identify Novel Biomarker Signatures"..... or maybe molecular fingerprint?

Major comment:

All raw and more useful processed data should be made freely accessible online. As should any code used to carry out the analysis. This represents an enormous tool/resource to the field.

Minor comments

Line 38: Change to "model. This was previously"

Line 76: delete "to" from and to gain insights

Line 80: include the stages of disease progression and time point in brackets

Methods:

-Pig biofluid collection – could refer to figure 1

-Why was blood collected in this rather invasive method rather than venopuncture? If venopuncture would work its would be worth highlighting it in the manuscript as an alternative sampling route.

-When detailing time points collected it would be advantageous to say pre clinical, mid and late (or something similar) in brackets beside the time points in order to provide the reader unfamiliar with CLN3 with a feel for the implications of the age. I dont recall seeing that the rational for time point selection was stated in the manuscript?

Proteomic methods

-Confirm no albumin depletion. If so, the coverage is excellent and requires a mention.

-Line 170: Sample pooling carried out - 250ul from each sample, then 40ul etc – how much protein was in each sample i.e. do we know that the same quantity of protein was in each pig serum sample for pooling. If not then a line to justify this and/or highlight quant per sample pre mix is an additional step others may wish to include when following this as a template for biomarker discovery.

Proteomic data analysis

-Normalisation – what happened to CLN3 protein levels before and after normalisation? This could be an excellent internal control if it was identified by MS.

-Supp fig 2. The point of this figure is somewhat unclear. It seems to show a decrease in batch effect but increase in variance in other replicates. Is that correct? possibly a rework of the figure legend to hand hold the reader through would be useful here.

Results

Line 301 What are the figures "balanced error rate (0.004 ± 0.010)" median/mean \pm SD or SEM?

-Line 333: I would like to see more information detailing that distinct clusters could be attributed to each time point.

Otherwise this only appears to be mentioned in the legend of fig 2c. I dont understand what is meant by enrichment in this context. Needs a bit of explanation/detail.

-Line 335: specific uniprot keywords were selected – why were these selected? Possibly on the basis of enrichment score? It would be good to understand the rational either to demonstrate that this is not cherry picking known features or explicit about why they were chose over other potential factors.

-Line 354: "Together this enabled the deepest Batten disease as well as porcine serum profile" I would suggest that this is not the case for batten disease in general for each of the individual techniques. Other manuscripts have identified higher protein coverage from MS using different tissues. Maybe limit the statement to indicate it is likely one of the most comprehensive combined -omics profiles of porcine serum.

-Line 360: This may be better displayed as a histogram with all cellular components on x axis and the bars coloured in, with time point representation so it is clear that some not all cellular components are seen at all time points and where they are the percentage changes with time.

-Line 387: change rebound – suggest "return towards WT levels" as an alternative.

-Line 398: I find this figure confusing. I am unsure about what this adds to the flow.

Discussion

-Line 418: change dearth to scarcity

Line 443-4: I would add this was observed from mid to late point disease only.

Line 454-456: THIS IS EXCELLENT - "Importantly, our data suggests that some of the molecular patterns are temporally restricted, suggesting that an ideal therapeutic strategy could potentially be tailored to an individual's specific biomarker signature of disease progression." But change data suggests to indicates.

-Line 458: A reference may be required here as I have seen most of these in our conventional MS data sets. But it is correct, and the point should be made that if we wanted to be certain of seeing these as part of a larger panel then perhaps targeted MS methodologies or other assays would be required.

-Line 473; include that NF-L is associated with late stage disease pathology

-Line 480: If NF-L included it could only indicate advanced pathology so worth considering that it may be too late in therapeutic window. But perhaps that is a larger question for a review nthan discussion here.

- It would be good to include a sentence that showed proteins of interest for other NDs were not significantly altered in CLN3. This then further strengthens the case that an unbiased approach is vital. (supplementary figure 1).

Figures

Fig 1.

-I would like to see pre mid and late symptomatic included in pig section of diagram either with words or an arrow at bottom showing the direction of increasing symptoms. Link back to text in the manuscript explaining the time point selection.

Fig 2.

-Line 677: relating to the enrichment of keyword, please include and expand on those shown in boxes in heatmap C.

-Not sure why NF-L is highlighted in fig 2b as seems to be dismissed in line 68-69 of manuscript.

-Heatmap boxes – there is no mention on why or how they became keywords

-Heat map – difficult to read. Not sure FDR is necessary here, so colour coding could be better just attributed to enrichment? The more enriched the brighter the colour maybe?

-This whole figure needs to be bigger. Suggest a whole page.

Fig 3.

-Panel B-H need explanation in the legend.

Fig 4.

- Difficult to read with colour coding. This could be better with a darker grey of things not detected in the dataset

Supp Fig.2

-I don't quite understand this. Does that mean decrease variance between batches but increased variance between all other sources of variance? Colours too similar for me to distinguish easily.

Supp Fig 3.

-Is this required?

Suppl table 3 and 4

-Could be better explained/more easily interpreted by others as a histogram with all components on x axis and the bars colour coded with % association with each time point stacked within bar.

Finally it may be nice to have a figure included showing a heatmap or something of the key proteins and metabolites detected at each stage of disease progression.

e.g something like this

6 24 36

CTD * ** ***

Version 1:

Reviewer comments:

Reviewer #1

(Remarks to the Author)

The current improved version answered all of my previous concerns. Congrats to the authors, your article is an important contribution for the lysosomal storage disease field.

Reviewer #3

(Remarks to the Author)

I appreciate the clear and concise way the reviewers have addressed all of my original comments. I understand the amount of time and effort which will have gone into doing so. It is also heartening that the authors did not shy away from any of the limitations associated with individual techniques employed during the research. Thanks for sharing this interesting and important work.

I have also specifically checked the response to ref 2s comments and they look adequate to me.

Please make the data from the 48h time point in the following query (pasted below) available in the data repository with the rest when published, as although not highly powered if the variability is low others may want to use the data themselves. It could be mentioned in the methods section that the data was produced but not included in the analysis because of xxx.

"The study initially comprised 4 timepoints (6, 24, 36 and 48-months) but the 48-month timepoint was dropped from the study "due to limited biological variance". This merits further explanation especially as the 48-month time-point, which presumably reflects late-stage disease, could be highly informative as changes in biomarkers that may be missed at earlier ages might be expected to be more pronounced as disease progresses.

This statement was an oversight that should have been caught during internal review. The 48- month time point was excluded from downstream analyses as we were not properly powered from a statistical standpoint (n=3/group). This was clarified in the text."

Thank you for all of these great suggestions. They have been very helpful in tightening up the manuscript and improving readability and interpretability for readers. We have addressed each of your critiques with edits to the manuscript as described below.

Reviewers' comments:

Reviewer #1 (Remarks to the Author):

The authors have presented a valuable contribution to the field by conducting a comprehensive multi-omics analysis of a CLN3 Δ ex7/8 minipig model. The methodological approach is sound, and the generated data provide a rich resource for future investigations. To enhance the impact and clarity of the manuscript, some aspects should be addressed:

Major Considerations:

Title: While the current title is informative, it might be slightly overstated. Consider revising it to better reflect the scope and depth of the findings. To keep it in its current form human validations are mandatory.

We appreciate this suggestion. We have updated the title to stress that the novel biomarker signatures were found in a model of Batten Disease.

Comparison with Existing Literature: The authors should explicitly address how their findings align with or diverge from previous studies on Batten disease, particularly those employing omics approaches. A comparative figure illustrating the overlap and differences between their results and published data would be beneficial. For example, compare their findings to the transcriptomic data in PMID: 38853929 (where is transcriptomic data?), PMID: 32592935 (large dataset, most not detected in proteomics), PMID: 30771446, and the metabolomic data in PMID: 38195117 (Short list), PMID: 36212622 (from Brudvig's previous work), PMID: 20485751 (Cannot find on Wegner), PMID: 16239221.

Given the divergent results often observed between transcriptomics and proteomics data we hesitate to make comparisons with previously published transcriptomic studies. We did, however, add language to the discussion to better describe similarities and differences in proteomics and metabolomics data. ***which paragraph***

Causality Analysis: While the multi-block analysis was useful for reducing the dimensionality of the data, exploring causal relationships between the identified biomarkers is crucial for understanding disease mechanisms. Employing Bayesian or other probabilistic networks could provide valuable insights in this regard. Please build them. Some published models include PMID: 39288230, PMID: 39251098, PMID: 38579163.

Thank you for this excellent suggestion. We are cautious to not over-interpret pathway relationships between proteins and metabolites for reasons that have been discussed in recent review articles. We have built still built Bayesian networks for significant analytes at 6-, 24-, and

36-month timepoints and have added them to the supplement. (Supplementary Figs 5-7)

The sex of the animals was not mentioned. Are these findings valid for male and female Batten disease pigs?

Thank you for pointing this out. Our findings are valid for both male and female pigs. This concern was addressed on page 5 in the “Pig Biofluid Collection” section of the methods.

Minor Points:

IACUC Number: Please ensure that the IACUC number is included in the manuscript.

We have added the IACUC protocol number for Exemplar Genetics in the “Pig Biofluid Collection” section of the methods.

The increase in lysosomal enzyme activities observed in the study is likely due to TFEB-mediated transcription. Discussing this aspect and addressing why only certain cathepsins were affected, while sphingolipid metabolism enzymes remained relatively unchanged at late stages, would provide a more complete understanding of the findings. Refer to PMID: 19556463 for more information on TFEB-mediated transcription.

This is a good point and is something that is important to mention. We added that both possibilities stated – “as a consequence of increased lysosomal mass or as a compensatory mechanism for clearing storage material” – are likely downstream consequences of TFEB activation as observed in other LSDs. Additionally, we could not come up with a reasonable rationale as to why only certain cathepsins were affected, while sphingolipid metabolism enzymes remained relatively unchanged at later stages. This is a question that will require further investigation.

By addressing these points, the authors can significantly enhance the quality and impact of their manuscript, making it a valuable contribution to the field of Batten disease research.

Thank you for these valuable insights. We have attempted to address all of your concerns to the best of our ability.

Reviewer #2 (Remarks to the Author):

This study describes a detailed metabolomic and proteomic analysis of plasma from a pig model of juvenile neuronal ceroid lipofuscinosis. This appears to be a detailed and useful study but lack of access to the primary data is a major problem.

Major comments.

It is standard in the biological mass spectrometry field to make all raw, database search and result files

accessible to reviewers during manuscript submission and publicly accessible after publication. While the data analysis in this submission appears sound, the ability to review the primary data, in particular raw and normalized TMT reporter intensities for proteomic analysis and label-free quantitation for metabolomics, is absolutely needed for full evaluation. This problem would be easily corrected by data submission to a public repository, e.g. MassIVE and providing reviewer access.

Thank you for catching this. Both proteomics and metabolomics datasets have been uploaded to public repositories (MassIVE and MetaboLights, respectively). They can be located via the following links:

Metabolomics data will be available via study # MTBLS1104 and the following URL once made public: <https://www.ebi.ac.uk/metabolights/MTBLS1107>

Our proteomics dataset are currently available at the following URL: <https://massive.ucsd.edu/ProteoSAFe/dataset.jsp?task=c0856cd9532f47cfa86554f54ce91870>

The study initially comprised 4 timepoints (6, 24, 36 and 48-months) but the 48-month timepoint was dropped from the study “due to limited biological variance”. This merits further explanation especially as the 48-month time-point, which presumably reflects late-stage disease, could be highly informative as changes in biomarkers that may be missed at earlier ages might be expected to be more pronounced as disease progresses.

This statement was an oversight that should have been caught during internal review. The 48-month time point was excluded from downstream analyses as we were not properly powered from a statistical standpoint (n=3/group). This was clarified in the text.

In the abstract, it is mentioned that 769 metabolites and 2,634 proteins were quantified as an aggregate of all samples but before depth of coverage can be properly assessed, it would be important to know the average number of each per sample.

The average number of metabolites/proteins detected per sample is critical to assessment of depth of coverage. The average number of metabolites/proteins detected has been calculated for each time point and added to the first paragraph of the results. Clarifying text is as follows:

“An average of 2398, 2438, 2431 proteins and 738, 747, 742 metabolites were detected in 6-, 24-, and 36-month WT animals, respectively, and 2430, 2437, 2439 proteins and 729, 744, 741 metabolites were detected in 6-, 24-, and 36-month *CLN3^{Δex7-8}* animals, respectively.”

Along similar lines, on P16, there is discussion attributing the depth of proteome coverage to be due to the nanoparticle digestion approach but it is not possible to determine what degree of improvement comes from this method versus the two-dimensional liquid chromatography. Additional comparison studies would really be needed to justify such conclusions.

Although you are certainly correct in that the depth of proteome coverage cannot be properly assessed without additional comparisons, we have added references to studies where similar comparisons have been made.

Also, cathepsins S and B are readily detectable in plasma using standard proteomics workflow e.g. data-independent acquisition.

Thank you for pointing this out. We have backed off a bit in the text and state that CTSS and CTSB are sometimes not detectable in plasma using standard proteomics workflow. We have added appropriate references at the end of this statement.

Minor comments.

P3. I agree with the authors that additional biomarker in JNCL may be valuable but the comments downplaying NFL because of “highly-variable elevations” probably needs further explanation to justify.

Although NFL demonstrates robust upregulation in late-stage disease, the data spread overlaps with that of healthy controls (Dang Do et. al., 2020). Clarifying text was added to justify this statement.

Fig 3. Is NFL measured by immunoassay or mass spectrometry ? This should be clarified in the legend and if former, comments re. detection in low abundance range by mass spectrometry are not relevant.

This is important to point out. NFL was quantified via Simoa targeted Neurology 4-PlexA assay; this was clarified in the figure legend. Additionally, NFL was removed from the statement discussing proteins in the low abundance range.

Reviewer #3 (Remarks to the Author):

This manuscript from Rechtzigel and colleagues under the guidance of Brudvig is a tour de force in terms of new data generation and handling for the field of lysosomal storage disorders. With a focus on CLN3, this could represent the beginnings of a more standardised pipeline for the generation of a molecular fingerprint for identifying, predicting and monitoring neurodegenerative disease progression in general. Both for tracking normal disease course and potentially in response to therapeutic intervention.

This also crucially frames the important role that larger animal model systems can play in discovery driven science with a view to support and play NDA enabling translational studies. Lysosomal storage disorders are a significant proportion of the "effective" therapies for neurodegenerative conditions. The availability of Brineuria for cln2 from Biomarin (via a canine model) and the permissions for gene therapy for CLN5 built on livestock models for scale up, efficacy and tox simultaneously, places the use of livestock in a more prominent role in the therapeutic development pipeline (especially as an alternative to NHP tox studies).

I believe that the depth of analysis here and the approaches provided will be of significant interest for researchers, clinicians and industrial stakeholders. but I have some queries and suggestions which I would like to see in order to clarify some of the methodology and processes presented here:

Firstly, the title could/should be changed. It is quite long and maybe fails to convey that this work makes use of novel resources like the CLN3 porcine model, AND that the approach could be replicated for any other neurodegenerative condition where such a model is available. I suggest shortening to something

like:

"Longitudinal Deep Multi-Omics Profiling of CLN3 Δ ex7/8 Minipig Model as a pipeline to identify Novel Biomarker Signatures"..... or maybe molecular fingerprint?

We appreciate your suggestion and agree that these options are much more fitting for this publication. The title has been changed to "Longitudinal Deep Multi-Omics Profiling in a CLN3 Δ ex7/8 Minipig Model as a Pipeline to Identify Novel Biomarker Signatures"

Major comment:

All raw and more useful processed data should be made freely accessible online. As should any code used to carry out the analysis. This represents an enormous tool/resource to the field.

Thank you for catching this. Raw metabolomics and proteomics datafiles have been uploaded to MetaboLights and MASSive, respectively, and are now publicly accessible.

Metabolomics data will be available via study # MTBLS1104 and the following URL once made public (having minor technical issues with MetaboLights but will make public ASAP):
<https://www.ebi.ac.uk/metabolights/MTBLS1107>

Our proteomics dataset is currently available at the following URL:
<https://massive.ucsd.edu/ProteoSAFe/dataset.jsp?task=c0856cd9532f47cfa86554f54ce91870>

Minor comments

Line 38: Change to "model. This was previously"

Line 76: delete "to" from and to gain insights

We appreciate these suggestions. Both suggestions were addressed in the text.

Line 80: include the stages of disease progression and time point in brackets

Stages of disease progression were added the "Pig Biofluid Collection" (6mo – pre-symptomatic, 24mo – mid-stage/symptomatic, 36mo – late-stage) section of the methods, and have been edited consistently throughout the text

Methods:

-Pig biofluid collection – could refer to figure 1

Time points descriptions were added to figure 1. Additionally, we have added a reference to figure 1 in the "Pig Biofluid Collection" section of the methods.

-Why was blood collected in this rather invasive method rather than venopuncture? If venopuncture would work its would be worth highlighting it in the manuscript as an alternative sampling route.

While venipuncture would be a preferable method in most circumstances, blood was collected at terminal tissue collection as these pigs were part of a larger characterization study for which they had met their pathology timepoint. This was addressed in the “Pig Biofluid Collection” section of the methods.

-When detailing time points collected it would be advantageous to say pre clinical, mid and late (or something similar) in brackets beside the time points in order to provide the reader unfamiliar with CLN3 with a feel for the implications of the age. I dont recall seeing that the rational for time point selection was stated in the manuscript?

You are correct in that we had not previously called this out in the text and would be very useful to readers not familiar with Batten Disease progression. We have addressed this in the text as follows: 6mo – pre-symptomatic, 24mo – mid-stage/symptomatic, 36mo – late-stage

Proteomic methods

-Confirm no albumin depletion. If so, the coverage is excellent and requires a mention.

Yes, we can confirm that we did not perform albumin depletion. We have clarified this in the methods section.

-Line 170: Sample pooling carried out - 250ul from each sample, then 40ul etc – how much protein was in each sample i.e. do we know that the same quantity of protein was in each pig serum sample for pooling. If not then a line to justify this and/or highlight quant per sample pre mix is an additional step others may wish to include when following this as a template for biomarker discovery.

This is important to clarify. We only used the pooling of 48-month samples for normalization across TMT batches and excluded in downstream analyses, so they were not used to ‘discover biomarkers’.

Proteomic data analysis

-Normalisation – what happened to CLN3 protein levels before and after normalisation? This could be an excellent internal control if it was identified by MS.

This is a great idea and would serve as a fantastic internal control. Unfortunately we were unable to do so as CLN3 was not detected in any of our samples.

-Supp fig 2. The point of this figure is somewhat unclear. It seems to show a decrease in batch effect but increase in variance in other replicates. Is that correct? possibly a rework of the figure legend to hand hold the reader through would be useful here.

While this is correct, the minor increase in variance across replicates is tolerable given the profound decrease in variance between TMT runs. We have now specified the figure legend that “mixture” refers to TMT run.

Results

Line 301 What are the figures "balanced error rate (0.004 ± 0.010)" median/mean \pm SD or SEM?

Thank you for inquiring. The balanced error rate is \pm SD and was clarified in the text.

-Line 333: I would like to see more information detailing that distinct clusters could be attributed to each time point. Otherwise this only appears to be mentioned in the legend of fig 2c. I don't understand what is meant by enrichment in this context. Needs a bit of explanation/detail.

To clarify, the U-map was used to generate clusters using combined datasets from all timepoints. We used functional enrichment analyses later in the manuscript to look at differences across timepoints.

-Line 335: specific uniprot keywords were selected – why were these selected? Possibly on the basis of enrichment score? It would be good to understand the rationale either to demonstrate that this is not cherry picking known features or explicit about why they were chosen over other potential factors.

Thank you for pointing this out as our rationale for selecting these keywords should certainly be clarified. The Uniprot keywords that were selected have a clear rationale that connects them to disease biology (muscle protein, thiol protease, cytokine/chemotaxis). This rationale was clarified in the text.

-Line 354: "Together this enabled the deepest Batten disease as well as porcine serum profile" I would suggest that this is not the case for batten disease in general for each of the individual techniques. Other manuscripts have identified higher protein coverage from MS using different tissues. Maybe limit the statement to indicate it is likely one of the most comprehensive combined -omics profiles of porcine serum.

Thank you for pointing this out. This sentence has been reworded to clarify that this was likely one of the most comprehensive profiles of a CLN3 disease model, and the deepest omics-profile of a porcine model of disease:

"Together, this enabled one of the most comprehensive -omics profiles of a CLN3 model, and the deepest omics-profile of porcine serum to date, quantifying novel putative disease biomarkers with a depth of coverage not typically achievable with standard serum proteomics workflows"

-Line 360: This may be better displayed as a histogram with all cellular components on x axis and the bars coloured in, with time point representation so it is clear that some not all cellular components are seen at all time points and where they are the percentage changes with time.

This is a great suggestion and is a much nicer representation of the data over time. Supplementary Tables 3 and 4 have been replaced with histograms representative of the same data (Supplementary figures 3 and 4).

-Line 387: change rebound – suggest "return towards WT levels" as an alternative.

Thank you for the suggestion, “return towards WT levels” is much more descriptive. This has been addressed in the text.

-Line 398: I find this figure confusing. I am unsure about what this adds to the flow.

Thank you for pointing this out. We have added text prior to the first mention of this figure to explain the rationale as to why it is currently included. For further clarification, sPLS-DA is similar to PCA but with more supervision. We have also added appropriate references to this section that the reader can refer to if they have questions.

Discussion

-Line 418: change dearth to scarcity

Line 443-4: I would add this was observed from mid to late point disease only.

Line 454-456: THIS IS EXCELLENT - "Importantly, our data suggests that some of the molecular patterns are temporally restricted, suggesting that an ideal therapeutic strategy could potentially be tailored to an individual’s specific biomarker signature of disease progression." But change data suggests to indicates.

Thank you for these suggestions to the text. These have all been addressed.

-Line 458: A reference may be required here as I have seen most of these in our conventional MS data sets. But it is correct, and the point should be made that if we wanted to be certain of seeing these as part of a larger panel then perhaps targeted MS methodologies or other assays would be required.

Thank you for catching this. We have adjusted the text stating that many of these proteins would be difficult to detect with a conventional unbiased proteomics workflow. Additionally, we have added references that have made appropriate comparisons.

-Line 473; include that NF-L is associated with late stage disease pathology

This is important to point out and was addressed in the text via the following statement:

“Associated with late-stage disease (i.e., neurodegeneration), one existing marker, NFL, offers the benefit of being closely linked to the central neuronal pathology...”

-Line 480: If NF-L included it could only indicate advanced pathology so worth considering that it may be too late in therapeutic window. But perhaps that is a larger question for a review than discussion here.

We appreciate the suggestion and agree that this is a better question for a review. We did add some additional language to emphasize that this marker is only helpful for later stage disease.

- It would be good to include a sentence that showed proteins of interest for other NDs were not significantly altered in CLN3. This then further strengthens the case that an unbiased approach is vital. (supplementary figure 1).

Thank you for the great suggestion. We have added text to point out that other established markers of NDs (TAU, GFAP, UCHL1) were not altered in our samples, strengthening the importance of an unbiased approach.

Figures

Fig 1.

-I would like to see pre mid and late symptomatic included in pig section of diagram either with words or an arrow at bottom showing the direction of increasing symptoms. Link back to text in the manuscript explaining the time point selection.

Figure 1 has been edited and an arrow with the three timepoints representing disease progression was added.

Fig 2.

-Line 677: relating to the enrichment of keyword, please include and expand on those shown in boxes in heatmap C.

-Not sure why NF-L is highlighted in fig 2b as seems to be dismissed in line 68-69 of manuscript.

-Heatmap boxes – there is no mention on why or how they became keywords

-Heat map – difficult to read. Not sure FDR is necessary here, so colour coding could be better just attributed to enrichment? The more enriched the brighter the colour maybe?

-This whole figure needs to be bigger. Suggest a whole page.

We appreciate your input regarding figure 2. Regarding keywords highlighted in boxes we have stated “Enrichments of keywords connected to disease biology (shown in boxes) are represented in...” as an explanation as to why we chose to highlight them. Additionally, the figure has been updated so that NF-L is no longer highlighted, and the color coding in Fig 2C has been adjusted so that it represents enrichment score as opposed to FDR. Lastly, this figure has been expanded to an entire page to improve readability.

Fig 3.

-Panel B-H need explanation in the legend.

We clarified in the figure legend that these were top biomarker candidates that we selected and discussed in detail in the manuscript text. Additional details have been added in reference to panel H.

Fig 4.

- Difficult to read with colour coding. This could be better with a darker grey of things not detected in the dataset

Thank you for catching this, the text in this figure was a bit difficult to read. The text color has been changed to black to alleviate this issue.

Supp Fig.2

-I don't quite understand this. Does that mean decrease variance between batches but increased variance between all other sources of variance? Colours too similar for me to distinguish easily.

Although this method of normalization does increase variance from other sources, we chose to normalize to all time points as it reduces the between-mixture variance minimizing the batch effect. This rationale was added to the end of the figure legend.

Supp Fig 3.

-Is this required?

Supplementary Figure 3 is certainly not required and has been removed from the supplementary materials.

Suppl table 3 and 4

-Could be better explained/more easily interpreted by others as a histogram with all components on x axis and the bars colour coded with % association with each time point stacked within bar.

As mentioned above, these tables have been replaced with histograms to better represent changes across timepoints.

Finally it may be nice to have a figure included showing a heatmap or something of the key proteins and metabolites detected at each stage of disease progression.

e.g something like this

6 24 36

CTD * * * * *

Thank you for this suggestion. We have added heatmaps (Supplementary Figs 3-4) of all significantly dysregulated proteins and metabolites that are shared by at least two of the three timepoints

Thank you for all of these great suggestions. They have been very helpful in tightening up the manuscript and improving readability and interpretability for readers. We have addressed each of your critiques with edits to the manuscript as described below.

Reviewers' comments:

Reviewer #1 (Remarks to the Author):

The authors have presented a valuable contribution to the field by conducting a comprehensive multi-omics analysis of a CLN3 Δ ex7/8 minipig model. The methodological approach is sound, and the generated data provide a rich resource for future investigations. To enhance the impact and clarity of the manuscript, some aspects should be addressed:

Major Considerations:

Title: While the current title is informative, it might be slightly overstated. Consider revising it to better reflect the scope and depth of the findings. To keep it in its current form human validations are mandatory.

We appreciate this suggestion. We have updated the title to stress that the novel biomarker signatures were found in a model of Batten Disease.

Comparison with Existing Literature: The authors should explicitly address how their findings align with or diverge from previous studies on Batten disease, particularly those employing omics approaches. A comparative figure illustrating the overlap and differences between their results and published data would be beneficial. For example, compare their findings to the transcriptomic data in PMID: 38853929 (where is transcriptomic data?), PMID: 32592935 (large dataset, most not detected in proteomics), PMID: 30771446, and the metabolomic data in PMID: 38195117 (Short list), PMID: 36212622 (from Brudvig's previous work), PMID: 20485751 (Cannot find on Wegner), PMID: 16239221.

Given the divergent results often observed between transcriptomics and proteomics data we hesitate to make comparisons with previously published transcriptomic studies. We did, however, add language to the discussion to better describe similarities and differences in proteomics and metabolomics data.

Causality Analysis: While the multi-block analysis was useful for reducing the dimensionality of the data, exploring causal relationships between the identified biomarkers is crucial for understanding disease mechanisms. Employing Bayesian or other probabilistic networks could provide valuable insights in this regard. Please build them. Some published models include PMID: 39288230, PMID: 39251098, PMID: 38579163.

Thank you for this excellent suggestion. We are cautious to not over-interpret pathway relationships between proteins and metabolites for reasons that have been discussed in recent review articles. We have built still built Bayesian networks for significant analytes at 6-, 24-, and

36-month timepoints and have added them to the supplement. (Supplementary Figs 5-7)

The sex of the animals was not mentioned. Are these findings valid for male and female Batten disease pigs?

Thank you for pointing this out. Our findings are valid for both male and female pigs. This concern was addressed on page 5 in the “Pig Biofluid Collection” section of the methods.

Minor Points:

IACUC Number: Please ensure that the IACUC number is included in the manuscript.

We have added the IACUC protocol number for Exemplar Genetics in the “Pig Biofluid Collection” section of the methods.

The increase in lysosomal enzyme activities observed in the study is likely due to TFEB-mediated transcription. Discussing this aspect and addressing why only certain cathepsins were affected, while sphingolipid metabolism enzymes remained relatively unchanged at late stages, would provide a more complete understanding of the findings. Refer to PMID: 19556463 for more information on TFEB-mediated transcription.

This is a good point and is something that is important to mention. We added that both possibilities stated – “as a consequence of increased lysosomal mass or as a compensatory mechanism for clearing storage material” – are likely downstream consequences of TFEB activation as observed in other LSDs. Additionally, we could not come up with a reasonable rationale as to why only certain cathepsins were affected, while sphingolipid metabolism enzymes remained relatively unchanged at later stages. This is a question that will require further investigation.

By addressing these points, the authors can significantly enhance the quality and impact of their manuscript, making it a valuable contribution to the field of Batten disease research.

Thank you for these valuable insights. We have attempted to address all of your concerns to the best of our ability.

Reviewer #2 (Remarks to the Author):

This study describes a detailed metabolomic and proteomic analysis of plasma from a pig model of juvenile neuronal ceroid lipofuscinosis. This appears to be a detailed and useful study but lack of access to the primary data is a major problem.

Major comments.

It is standard in the biological mass spectrometry field to make all raw, database search and result files

accessible to reviewers during manuscript submission and publicly accessible after publication. While the data analysis in this submission appears sound, the ability to review the primary data, in particular raw and normalized TMT reporter intensities for proteomic analysis and label-free quantitation for metabolomics, is absolutely needed for full evaluation. This problem would be easily corrected by data submission to a public repository, e.g. MassIVE and providing reviewer access.

Thank you for catching this. Both proteomics and metabolomics datasets have been uploaded to public repositories (MassIVE and MetaboLights, respectively). They can be located via the following links:

Metabolomics data will be available via study # MTBLS1104 and the following URL once made public: <https://www.ebi.ac.uk/metabolights/MTBLS1107>

Our proteomics dataset are currently available at the following URL: <https://massive.ucsd.edu/ProteoSAFe/dataset.jsp?task=c0856cd9532f47cfa86554f54ce91870>

The study initially comprised 4 timepoints (6, 24, 36 and 48-months) but the 48-month timepoint was dropped from the study “due to limited biological variance”. This merits further explanation especially as the 48-month time-point, which presumably reflects late-stage disease, could be highly informative as changes in biomarkers that may be missed at earlier ages might be expected to be more pronounced as disease progresses.

This statement was an oversight that should have been caught during internal review. The 48-month time point was excluded from downstream analyses as we were not properly powered from a statistical standpoint (n=3/group). This was clarified in the text.

In the abstract, it is mentioned that 769 metabolites and 2,634 proteins were quantified as an aggregate of all samples but before depth of coverage can be properly assessed, it would be important to know the average number of each per sample.

The average number of metabolites/proteins detected per sample is critical to assessment of depth of coverage. The average number of metabolites/proteins detected has been calculated for each time point and added to the first paragraph of the results. Clarifying text is as follows:

“An average of 2398, 2438, 2431 proteins and 738, 747, 742 metabolites were detected in 6-, 24-, and 36-month WT animals, respectively, and 2430, 2437, 2439 proteins and 729, 744, 741 metabolites were detected in 6-, 24-, and 36-month *CLN3*^{Δex7-8} animals, respectively.”

Along similar lines, on P16, there is discussion attributing the depth of proteome coverage to be due to the nanoparticle digestion approach but it is not possible to determine what degree of improvement comes from this method versus the two-dimensional liquid chromatography. Additional comparison studies would really be needed to justify such conclusions.

Although you are certainly correct in that the depth of proteome coverage cannot be properly assessed without additional comparisons, we have added references to studies where similar comparisons have been made.

Also, cathepsins S and B are readily detectable in plasma using standard proteomics workflow e.g. data-independent acquisition.

Thank you for pointing this out. We have backed off a bit in the text and state that CTSS and CTSB are sometimes not detectable in plasma using standard proteomics workflow. We have added appropriate references at the end of this statement.

Minor comments.

P3. I agree with the authors that additional biomarker in JNCL may be valuable but the comments downplaying NFL because of “highly-variable elevations” probably needs further explanation to justify.

Although NFL demonstrates robust upregulation in late-stage disease, the data spread overlaps with that of healthy controls (Dang Do et. al., 2020). Clarifying text was added to justify this statement.

Fig 3. Is NFL measured by immunoassay or mass spectrometry ? This should be clarified in the legend and if former, comments re. detection in low abundance range by mass spectrometry are not relevant.

This is important to point out. NFL was quantified via Simoa targeted Neurology 4-PlexA assay; this was clarified in the figure legend. Additionally, NFL was removed from the statement discussing proteins in the low abundance range.

Reviewer #3 (Remarks to the Author):

This manuscript from Rechtzigel and colleagues under the guidance of Brudvig is a tour de force in terms of new data generation and handling for the field of lysosomal storage disorders. With a focus on CLN3, this could represent the beginnings of a more standardised pipeline for the generation of a molecular fingerprint for identifying, predicting and monitoring neurodegenerative disease progression in general. Both for tracking normal disease course and potentially in response to therapeutic intervention.

This also crucially frames the important role that larger animal model systems can play in discovery driven science with a view to support and play NDA enabling translational studies. Lysosomal storage disorders are a significant proportion of the "effective" therapies for neurodegenerative conditions. The availability of Brineuria for cln2 from Biomarin (via a canine model) and the permissions for gene therapy for CLN5 built on livestock models for scale up, efficacy and tox simultaneously, places the use of livestock in a more prominent role in the therapeutic development pipeline (especially as an alternative to NHP tox studies).

I believe that the depth of analysis here and the approaches provided will be of significant interest for researchers, clinicians and industrial stakeholders. but I have some queries and suggestions which I would like to see in order to clarify some of the methodology and processes presented here:

Firstly, the title could/should be changed. It is quite long and maybe fails to convey that this work makes use of novel resources like the CLN3 porcine model, AND that the approach could be replicated for any other neurodegenerative condition where such a model is available. I suggest shortening to something

like:

"Longitudinal Deep Multi-Omics Profiling of CLN3 Δ ex7/8 Minipig Model as a pipeline to identify Novel Biomarker Signatures"..... or maybe molecular fingerprint?

We appreciate your suggestion and agree that these options are much more fitting for this publication. The title has been changed to "Longitudinal Deep Multi-Omics Profiling in a CLN3 Δ ex7/8 Minipig Model as a Pipeline to Identify Novel Biomarker Signatures"

Major comment:

All raw and more useful processed data should be made freely accessible online. As should any code used to carry out the analysis. This represents an enormous tool/resource to the field.

Thank you for catching this. Raw metabolomics and proteomics datafiles have been uploaded to MetaboLights and MASSive, respectively, and are now publicly accessible.

Metabolomics data will be available via study # MTBLS1104 and the following URL once made public (having minor technical issues with MetaboLights but will make public ASAP):
<https://www.ebi.ac.uk/metabolights/MTBLS1107>

Our proteomics dataset is currently available at the following URL:
<https://massive.ucsd.edu/ProteoSAFe/dataset.jsp?task=c0856cd9532f47cfa86554f54ce91870>

Minor comments

Line 38: Change to "model. This was previously"

Line 76: delete "to" from and to gain insights

We appreciate these suggestions. Both suggestions were addressed in the text.

Line 80: include the stages of disease progression and time point in brackets

Stages of disease progression were added the "Pig Biofluid Collection" (6mo – pre-symptomatic, 24mo – mid-stage/symptomatic, 36mo – late-stage) section of the methods, and have been edited consistently throughout the text

Methods:

-Pig biofluid collection – could refer to figure 1

Time points descriptions were added to figure 1. Additionally, we have added a reference to figure 1 in the "Pig Biofluid Collection" section of the methods.

-Why was blood collected in this rather invasive method rather than venopuncture? If venopuncture would work its would be worth highlighting it in the manuscript as an alternative sampling route.

While venipuncture would be a preferable method in most circumstances, blood was collected at terminal tissue collection as these pigs were part of a larger characterization study for which they had met their pathology timepoint. This was addressed in the “Pig Biofluid Collection” section of the methods.

-When detailing time points collected it would be advantageous to say pre clinical, mid and late (or something similar) in brackets beside the time points in order to provide the reader unfamiliar with CLN3 with a feel for the implications of the age. I dont recall seeing that the rational for time point selection was stated in the manuscript?

You are correct in that we had not previously called this out in the text and would be very useful to readers not familiar with Batten Disease progression. We have addressed this in the text as follows: 6mo – pre-symptomatic, 24mo – mid-stage/symptomatic, 36mo – late-stage

Proteomic methods

-Confirm no albumin depletion. If so, the coverage is excellent and requires a mention.

Yes, we can confirm that we did not perform albumin depletion. We have clarified this in the methods section.

-Line 170: Sample pooling carried out - 250ul from each sample, then 40ul etc – how much protein was in each sample i.e. do we know that the same quantity of protein was in each pig serum sample for pooling. If not then a line to justify this and/or highlight quant per sample pre mix is an additional step others may wish to include when following this as a template for biomarker discovery.

This is important to clarify. We only used the pooling of 48-month samples for normalization across TMT batches and excluded in downstream analyses, so they were not used to ‘discover biomarkers’.

Proteomic data analysis

-Normalisation – what happened to CLN3 protein levels before and after normalisation? This could be an excellent internal control if it was identified by MS.

This is a great idea and would serve as a fantastic internal control. Unfortunately we were unable to do so as CLN3 was not detected in any of our samples.

-Supp fig 2. The point of this figure is somewhat unclear. It seems to show a decrease in batch effect but increase in variance in other replicates. Is that correct? possibly a rework of the figure legend to hand hold the reader through would be useful here.

While this is correct, the minor increase in variance across replicates is tolerable given the profound decrease in variance between TMT runs. We have now specified the figure legend that “mixture” refers to TMT run.

Results

Line 301 What are the figures "balanced error rate (0.004 ± 0.010)" median/mean ± SD or SEM?

Thank you for inquiring. The balanced error rate is ± SD and was clarified in the text.

-Line 333: I would like to see more information detailing that distinct clusters could be attributed to each time point. Otherwise this only appears to be mentioned in the legend of fig 2c. I don't understand what is meant by enrichment in this context. Needs a bit of explanation/detail.

To clarify, the U-map was used to generate clusters using combined datasets from all timepoints. We used functional enrichment analyses later in the manuscript to look at differences across timepoints.

-Line 335: specific uniprot keywords were selected – why were these selected? Possibly on the basis of enrichment score? It would be good to understand the rationale either to demonstrate that this is not cherry picking known features or explicit about why they were chosen over other potential factors.

Thank you for pointing this out as our rationale for selecting these keywords should certainly be clarified. The Uniprot keywords that were selected have a clear rationale that connects them to disease biology (muscle protein, thiol protease, cytokine/chemotaxis). This rationale was clarified in the text.

-Line 354: "Together this enabled the deepest Batten disease as well as porcine serum profile" I would suggest that this is not the case for batten disease in general for each of the individual techniques. Other manuscripts have identified higher protein coverage from MS using different tissues. Maybe limit the statement to indicate it is likely one of the most comprehensive combined -omics profiles of porcine serum.

Thank you for pointing this out. This sentence has been reworded to clarify that this was likely one of the most comprehensive profiles of a CLN3 disease model, and the deepest omics-profile of a porcine model of disease:

"Together, this enabled one of the most comprehensive -omics profiles of a CLN3 model, and the deepest omics-profile of porcine serum to date, quantifying novel putative disease biomarkers with a depth of coverage not typically achievable with standard serum proteomics workflows"

-Line 360: This may be better displayed as a histogram with all cellular components on x axis and the bars coloured in, with time point representation so it is clear that some not all cellular components are seen at all time points and where they are the percentage changes with time.

This is a great suggestion and is a much nicer representation of the data over time. Supplementary Tables 3 and 4 have been replaced with histograms representative of the same data (Supplementary figures 3 and 4).

-Line 387: change rebound – suggest "return towards WT levels" as an alternative.

Thank you for the suggestion, “return towards WT levels” is much more descriptive. This has been addressed in the text.

-Line 398: I find this figure confusing. I am unsure about what this adds to the flow.

Thank you for pointing this out. We have added text prior to the first mention of this figure to explain the rationale as to why it is currently included. For further clarification, sPLS-DA is similar to PCA but with more supervision. We have also added appropriate references to this section that the reader can refer to if they have questions.

Discussion

-Line 418: change dearth to scarcity

Line 443-4: I would add this was observed from mid to late point disease only.

Line 454-456: THIS IS EXCELLENT - "Importantly, our data suggests that some of the molecular patterns are temporally restricted, suggesting that an ideal therapeutic strategy could potentially be tailored to an individual’s specific biomarker signature of disease progression." But change data suggests to indicates.

Thank you for these suggestions to the text. These have all been addressed.

-Line 458: A reference may be required here as I have seen most of these in our conventional MS data sets. But it is correct, and the point should be made that if we wanted to be certain of seeing these as part of a larger panel then perhaps targeted MS methodologies or other assays would be required.

Thank you for catching this. We have adjusted the text stating that many of these proteins would be difficult to detect with a conventional unbiased proteomics workflow. Additionally, we have added references that have made appropriate comparisons.

-Line 473; include that NF-L is associated with late stage disease pathology

This is important to point out and was addressed in the text via the following statement:

“Associated with late-stage disease (i.e., neurodegeneration), one existing marker, NFL, offers the benefit of being closely linked to the central neuronal pathology...”

-Line 480: If NF-L included it could only indicate advanced pathology so worth considering that it may be too late in therapeutic window. But perhaps that is a larger question for a review than discussion here.

We appreciate the suggestion and agree that this is a better question for a review. We did add some additional language to emphasize that this marker is only helpful for later stage disease.

- It would be good to include a sentence that showed proteins of interest for other NDs were not significantly altered in CLN3. This then further strengthens the case that an unbiased approach is vital. (supplementary figure 1).

Thank you for the great suggestion. We have added text to point out that other established markers of NDs (TAU, GFAP, UCHL1) were not altered in our samples, strengthening the importance of an unbiased approach.

Figures

Fig 1.

-I would like to see pre mid and late symptomatic included in pig section of diagram either with words or an arrow at bottom showing the direction of increasing symptoms. Link back to text in the manuscript explaining the time point selection.

Figure 1 has been edited and an arrow with the three timepoints representing disease progression was added.

Fig 2.

-Line 677: relating to the enrichment of keyword, please include and expand on those shown in boxes in heatmap C.

-Not sure why NF-L is highlighted in fig 2b as seems to be dismissed in line 68-69 of manuscript.

-Heatmap boxes – there is no mention on why or how they became keywords

-Heat map – difficult to read. Not sure FDR is necessary here, so colour coding could be better just attributed to enrichment? The more enriched the brighter the colour maybe?

-This whole figure needs to be bigger. Suggest a whole page.

We appreciate your input regarding figure 2. Regarding keywords highlighted in boxes we have stated “Enrichments of keywords connected to disease biology (shown in boxes) are represented in...” as an explanation as to why we chose to highlight them. Additionally, the figure has been updated so that NF-L is no longer highlighted, and the color coding in Fig 2C has been adjusted so that it represents enrichment score as opposed to FDR. Lastly, this figure has been expanded to an entire page to improve readability.

Fig 3.

-Panel B-H need explanation in the legend.

We clarified in the figure legend that these were top biomarker candidates that we selected and discussed in detail in the manuscript text. Additional details have been added in reference to panel H.

Fig 4.

- Difficult to read with colour coding. This could be better with a darker grey of things not detected in the dataset

Thank you for catching this, the text in this figure was a bit difficult to read. The text color has been changed to black to alleviate this issue.

Supp Fig.2

-I don't quite understand this. Does that mean decrease variance between batches but increased variance between all other sources of variance? Colours too similar for me to distinguish easily.

Although this method of normalization does increase variance from other sources, we chose to normalize to all time points as it reduces the between-mixture variance minimizing the batch effect. This rationale was added to the end of the figure legend.

Supp Fig 3.

-Is this required?

Supplementary Figure 3 is certainly not required and has been removed from the supplementary materials.

Suppl table 3 and 4

-Could be better explained/more easily interpreted by others as a histogram with all components on x axis and the bars colour coded with % association with each time point stacked within bar.

As mentioned above, these tables have been replaced with histograms to better represent changes across timepoints.

Finally it may be nice to have a figure included showing a heatmap or something of the key proteins and metabolites detected at each stage of disease progression.

e.g something like this

6 24 36

CTD * * * * *

Thank you for this suggestion. We have added heatmaps (Supplementary Figs 3-4) of all significantly dysregulated proteins and metabolites that are shared by at least two of the three timepoints

Referee Comments 04July2025

Reviewer #1 (Remarks to the Author):

The current improved version answered all of my previous concerns. Congrats to the authors, your article is an important contribution for the lysosomal storage disease field.

Reviewer #3 (Remarks to the Author):

I appreciate the clear and concise way the reviewers have addressed all of my original comments. I understand the amount of time and effort which will have gone into doing so. It is also heartening that the authors did not shy away from any of the limitations associated with individual techniques employed during the research. Thanks for sharing this interesting and important work.

I have also specifically checked the response to ref 2s comments and they look adequate to me.

Please make the data from the 48h time point in the following query (pasted below) available in the data repository with the rest when published, as although not highly powered if the variability is low others may want to use the data themselves. It could be mentioned in the methods section that the data was produced but not included in the analysis because of xxx.

Line 240-242 were edited as such: “Although data were produced, samples from 48-month-old subjects were subsequently excluded from downstream statistical analysis due to a lack of appropriate statistical power (n=3/group).”

We agree that it is important to make these data available to the community and will make sure that 48-month data are uploaded to appropriate repositories.

“The study initially comprised 4 timepoints (6, 24, 36 and 48-months) but the 48-month timepoint was dropped from the study “due to limited biological variance”. This merits further explanation especially as the 48-month time-point, which presumably reflects late-stage disease, could be highly informative as changes in biomarkers that may be missed at earlier ages might be expected to be more pronounced as disease progresses.

This statement was an oversight that should have been caught during internal review. The 48-month time point was excluded from downstream analyses as we were not properly powered from a statistical standpoint (n=3/group). This was clarified in the text.”